# TAMUNA: Doubly Accelerated Federated Learning with Local Training, Compression, and Partial Participation

## Abstract

In federated learning, a large number of users collaborate to learn a global model. They alternate local computations and communication with a distant server. Communication, which can be slow and costly, is the main bottleneck in this setting. In addition to communication-efficiency, a robust algorithm should allow for partial participation, the desirable feature that not all clients need to participate to every round of the training process. To reduce the communication load and therefore accelerate distributed gradient descent, two strategies are popular: 1) communicate less frequently; that is, perform several iterations of *local* computations between the communication rounds; and 2) communicate *compressed* information instead of full-dimensional vectors. We propose TAMUNA, the first algorithm for distributed optimization and federated learning, which harnesses these two strategies jointly and allows for partial participation. TAMUNA converges linearly to an exact solution in the strongly convex setting, with a doubly accelerated rate: it provably benefits from the two acceleration mechanisms provided by local training and compression, namely a better dependency on the condition number of the functions and on the model dimension, respectively.

## 1 Introduction

Federated Learning (FL) is a novel paradigm for training supervised machine learning models. Initiated a few years ago (Konečný et al., 2016a;b; McMahan et al., 2017; Bonawitz et al., 2017), it has become a rapidly growing interdisciplinary field. The key idea is to exploit the wealth of information stored on edge devices, such as mobile phones, sensors and hospital workstations, to train global models, in a collaborative way, while handling a multitude of challenges, like data privacy (Kairouz et al., 2021; Li et al., 2020a; Wang et al., 2021). In contrast to centralized learning in a datacenter, in FL, the parallel computing units have private data stored on each of them and communicate with a distant orchestrating server, which aggregates the information and synchronizes the computations, so that the process reaches a consensus and converges to a globally optimal model. In this framework, communication between the parallel workers and the server, which can take place over the internet or cell phone network, can be slow, costly, and unreliable. Thus, communication dominates the overall duration and cost of the process and is the main bottleneck to be addressed by the community, before FL can be widely adopted and applied in our daily lives.

The baseline algorithm of distributed Gradient Descent (GD) alternates between two steps: one round of parallel computation of the local function gradients at the current model estimate, and one round of communication of these gradient vectors to the server, which averages them to form the new estimate for the next iteration. To decrease the communication load, two strategies can be used: 1) communicate less frequently, or equivalently do more *local computations* between successive communication rounds; or 2) *compress* the communicated vectors. We detail these two strategies in Section 1.3. Moreover, in practical applications where FL is deployed, it is unrealistic to assume that all clients are available 100% of the time to perform the required computation and communication operations. Thus, *partial participation* is an essential feature in practice, whereby only part of the clients need to participate in any given round of the process, while maintaining the overall convergence guarantees.

In this paper, we propose a new randomized algorithm named TAMUNA, which combines local training and compression for communication-efficient FL. It is variance-reduced (Hanzely & Richtárik, 2019; Gorbunov et al., 2020a; Gower et al., 2020), so that it converges to an exact solution (with exact gradients), and provably benefits from the two mechanisms: the convergence rate is doubly accelerated, with a better dependency on the condition number of the functions and on the dimension of the model, in comparison with GD. In addition, TAMUNA handles partial participation of the clients. In the remainder of this section, we formulate the setup, we propose a new model to characterize the communication complexity, we present the state of the art, and we summarize our contributions.

## 1.1 Formalism

We consider a distributed client-server setting, in which $n \geq 2$ clients perform computations in parallel and communicate back and forth with a server. We study the convex optimization problem:

$$\underset{x \in \mathbb{R}^d}{\text{minimize}} \ f(x) \coloneqq \frac{1}{n} \sum_{i=1}^{n} f_i(x), \tag{1}$$

where each function $f_i : \mathbb{R}^d \to \mathbb{R}$ models the individual cost of client $i \in [n] \coloneqq \{1, \ldots, n\}$, based on its underlying private data. The number $n$ of clients, as well as the dimension $d \geq 1$ of the model, are typically large. This problem is of key importance as it is an abstraction of empirical risk minimization, the dominant framework in supervised machine learning.

For every $i \in [n]$, the function $f_i$ is supposed $L$-smooth and $\mu$-strongly convex,[1] for some $L \geq \mu > 0$ (a sublinear convergence result is derived in the Appendix for the merely convex case, i.e. $\mu = 0$). Thus, the sought solution $x^\star$ of (1) exists and is unique. We define $\kappa \coloneqq \frac{L}{\mu}$. We focus on the strongly convex case, because the analysis of linear convergence rates in this setting gives clear insights and allows us to deepen our theoretical understanding of the algorithmic mechanisms under study; in our case, local training, communication compression, and partial participation. The analysis of algorithms converging to a point with zero gradient in (1) with nonconvex functions relies on significantly different proof techniques (Karimireddy et al., 2021; Das et al., 2022), so the nonconvex setting is out of the scope of this paper.

To solve the problem (1), the baseline algorithm of Gradient Descent (GD) consists in the simple iteration, for $t = 0, 1, \ldots,$

$$x^{t+1} \coloneqq x^t - \frac{\gamma}{n} \sum_{i=1}^{n} \nabla f_i(x^t),$$

for some stepsize $\gamma \in (0, \frac{2}{L})$. That is, at iteration $t$, $x^t$ is first broadcast by the server to all clients, which compute the gradients $\nabla f_i(x^t)$ in parallel. These vectors are then sent to the server, which averages them and performs the gradient descent step. It is well known that for $\gamma = \Theta(\frac{1}{L})$, GD converges linearly, with iteration complexity $\mathcal{O}(\kappa \log \epsilon^{-1})$ to reach $\epsilon$-accuracy. Since $d$-dimensional vectors are communicated at every iteration, the communication complexity of GD in number of reals is $\mathcal{O}(d\kappa \log \epsilon^{-1})$. Our goal is a twofold acceleration of GD, with a better dependency to both $\kappa$ and $d$ in this complexity. We want to achieve this goal by leveraging the best of the two popular mechanisms of local training and communication compression.

## 1.2 Asymmetric communication regime

**Uplink and downlink communication**. We call *uplink communication* (UpCom) the parallel transmission of data from the clients to the server and *downlink communication* (DownCom) the broadcast of the same message from the server to all clients. UpCom is usually significantly slower than DownCom, just like uploading is slower than downloading on the internet or cell phone network. This can be due to the asymmetry of the service provider's systems or protocols used on the communication network, or cache memory and

---

[1] A function $f : \mathbb{R}^d \to \mathbb{R}$ is said to be $L$-smooth if it is differentiable and its gradient is Lipschitz continuous with constant $L$; that is, for every $x \in \mathbb{R}^d$ and $y \in \mathbb{R}^d$, $\|\nabla f(x) - \nabla f(y)\| \leq L\|x - y\|$, where, here and throughout the paper, the norm is the Euclidean norm. $f$ is said to be $\mu$-strongly convex if $f - \frac{\mu}{2}\|\cdot\|^2$ is convex. We refer to Bauschke & Combettes (2017) for such standard notions of convex analysis.

aggregation speed constraints of the server, which has to decode and average the large number $n$ of vectors received at the same time during UpCom.

**Communication complexity**. We measure the UpCom or DownCom complexity as the expected number of communication rounds needed to estimate a solution with $\epsilon$-accuracy, *multiplied by* the number of real values sent during a communication round between the server and any client. Thus, the UpCom or DownCom complexity of GD is $\mathcal{O}(d\kappa \log \epsilon^{-1})$). We leave it for future work to refine this model of counting real numbers, to take into account how sequences of real numbers are quantized into bitstreams, achieving further compression (Horváth et al., 2022; Albasyoni et al., 2020).

**A model for the overall communication complexity**. Since UpCom is usually slower than DownCom, we propose to measure the *total communication* (TotalCom) complexity as a weighted sum of the two UpCom and DownCom complexities: we assume that the UpCom cost is 1 (unit of time per transmitted real number), whereas the downCom cost is $\alpha \in [0, 1]$. Therefore,

$$\text{TotalCom} = \text{UpCom} + \alpha.\text{DownCom}. \tag{2}$$

A symmetric but unrealistic communication regime corresponds to $\alpha = 1$, whereas ignoring downCom and focusing on UpCom, which is usually the limiting factor, corresponds to $\alpha = 0$. We will provide explicit expressions of the parameters of our algorithm to minimize the TotalCom complexity for any given $\alpha \in [0, 1]$, keeping in mind that realistic settings correspond to small values of $\alpha$. Thus, our model of communication complexity is richer than only considering $\alpha = 0$, as is usually the case.

## 1.3 Communication efficiency in FL: state of the art

Two approaches come naturally to mind to decrease the communication load: *Local Training* (LT), which consists in communicating less frequently than at every iteration, and *Communication Compression* (CC), which consists in sending less than $d$ floats during every communication round. In this section, we review existing work related to these two strategies and to *Partial Participation* (PP).

### 1.3.1 Local Training (LT)

LT is a conceptually simple and surprisingly powerful communication-acceleration technique. It consists in the clients performing multiple local GD steps instead of only one, between successive communication rounds. This intuitively results in "better" information being communicated, so that less communication rounds are needed to reach a given accuracy. As shown by ample empirical evidence, LT is very efficient in practice. It was popularized by the FedAvg algorithm of McMahan et al. (2017), in which LT is a core component. However, LT was heuristic and no theory was provided in their paper. LT was analyzed in several works, in the homogeneous, or i.i.d. data, regime (Haddadpour & Mahdavi, 2019), and in the heterogeneous regime, which is more representative in FL (Khaled et al., 2019; Stich, 2019; Khaled et al., 2020; Li et al., 2020b; Woodworth et al., 2020; Gorbunov et al., 2021; Glasgow et al., 2022). It stands out that LT suffers from so-called client drift, which is the fact that the local model obtained by client $i$ after several local GD steps approaches the minimizer of its local cost function $f_i$. The discrepancy between the exact solution $x^\star$ of (1) and the approximate solution obtained at convergence of LT was characterized in Malinovsky et al. (2020). This deficiency of LT was corrected in the Scaffold algorithm of Karimireddy et al. (2020) by introducing control variates, which correct for the client drift, so that the algorithm converges linearly to the exact solution. S-Local-GD (Gorbunov et al., 2021) and FedLin (Mitra et al., 2021) were later proposed, with similar convergence properties. Yet, despite the empirical superiority of these recent algorithms relying on LT, their communication complexity remains the same as vanilla GD, i.e. $\mathcal{O}(d\kappa \log \epsilon^{-1})$.

Most recently, a breakthrough was made with the appearance of *accelerated* LT methods. Scaffnew, proposed by Mishchenko et al. (2022), is the first LT-based algorithm achieving $\mathcal{O}(d\sqrt{\kappa} \log \epsilon^{-1})$ accelerated communication complexity. In Scaffnew, communication is triggered randomly with a small probability $p$ at every iteration. Thus, the expected number of local GD steps between two communication rounds is $1/p$. By choosing $p = 1/\sqrt{\kappa}$, the optimal dependency on $\sqrt{\kappa}$ instead of $\kappa$ (Scaman et al., 2019) is obtained. In this paper, we propose to go even further and tackle the multiplicative factor $d$ in the complexity of Scaffnew.

Scaffnew has been extended in Malinovsky et al. (2022), using calls to variance-reduced (Gorbunov et al., 2020a; Gower et al., 2020) stochastic gradient estimates instead of exact gradients. It has also been analyzed in Condat & Richtárik (2023) as a particular case of RandProx, a primal-dual algorithm with a general randomized and variance-reduced dual update. Conceptually, TAMUNA is inspired by RandProx, with the dual update corresponding to the intermittent update of the control variates of the participating clients. TAMUNA is not a particular case of RandProx, though, because the primal update of the model and the dual update of the control variates are decoupled. Without compression and in case of full participation, TAMUNA reverts to Scaffnew.

A different approach was developed by Sadiev et al. (2022a) with the APDA-Inexact algorithm, and then by Grudzień et al. (2023) with the 5GCS algorithm: in both algorithms, the local steps correspond to an inner loop to compute a proximity operator inexactly.

### 1.3.2 Partial Participation (PP)

PP, a.k.a. client sampling, is the property that not all clients need to participate in a given round, consisting of a series of local steps followed by communication with the server. This is an important feature for a FL method, since in practice, there are many reasons for which a client might be idle and unable to do any computation and communication for a certain period of time. PP in SGD-type methods is now well understood (Gower et al., 2019; Condat & Richtárik, 2022), but its combination with LT has remained unconvincing so far. Scaffold allows for LT and PP, but its communication complexity does not benefit from LT. The variance-reduced FedVARP algorithm with LT and PP has been proposed Jhunjhunwala et al. (2022), for nonconvex problems and with a bounded global variance assumption that does not hold in our setting. Scaffnew does not allow for PP. This was the motivation for Grudzień et al. (2023) to develop 5GCS, which is, to the best of our knowledge, the first and only algorithm enabling LT and PP, and enjoying accelerated communication. We refer to Grudzień et al. (2023) for a detailed discussion of the literature of LT and PP. 5GCS is completely different from Scaffnew and based on Point-SAGA (Defazio, 2016) instead of GD. Thus, it is an indirect, or two-level, combination of LT and PP: PP comes from the random selection of the activated proximity operators, whereas LT corresponds to an inner loop to compute these proximity operators inexactly. TAMUNA is a direct combination of LT and PP as two intertwined stochastic processes. TAMUNA reverts to Scaffnew in case of full participation (and no compression); in other words, TAMUNA is the first generalization of Scaffnew to PP, and it fully retains its LT-based communication acceleration benefits.

Throughout the paper, we denote by $c \in \{2, \ldots, n\}$ the cohort size, or number of active clients participating in every round. We report in Table 1 the communication complexity of the two known algorithms converging linearly to the exact solution, while allowing for LT and PP, namely Scaffold and 5GCS. Scaffold is not accelerated, with a complexity depending on $\kappa$, and 5GCS is accelerated with respect to $\kappa$ but not $d$. Also, in 5GCS the number of local steps in each communication round is fixed of order at least $\left(\sqrt{\frac{c\kappa}{n}} + 1\right) \log \kappa$, whereas in TAMUNA it is random and typically much smaller, of order $\sqrt{\frac{s\kappa}{n}} + 1$, where $s$ can be as small as 2, see (14).

### 1.3.3 Communication Compression (CC)

To decrease the communication complexity, a widely used strategy is to make use of (lossy) compression; that is, a possibly randomized mapping $\mathcal{C} : \mathbb{R}^d \to \mathbb{R}^d$ is applied to the vector $x$ that needs to be communicated, with the property that it is much faster to transfer $\mathcal{C}(x)$ than the full $d$-dimensional vector $x$. A popular sparsifying compressor is rand-$k$, for some $k \in [d] \coloneqq \{1, \ldots, d\}$, which multiplies $k$ elements of $x$, chosen uniformly at random, by $d/k$, and sets the other ones to zero. If the receiver knows which coordinates have been selected, e.g. by running the same pseudo-random generator, only these $k$ elements of $x$ are actually communicated, so that the communication complexity is divided by the compression factor $d/k$. Another sparsifying compressor is top-$k$, which keeps the $k$ elements of $x$ with largest absolute values unchanged and sets the other ones to zero. Some compressors, like rand-$k$, are unbiased; that is, $\mathbb{E}[\mathcal{C}(x)] = x$ for every $x \in \mathbb{R}^d$, where $\mathbb{E}[\cdot]$ denotes the expectation. On the other hand, compressors like top-$k$ are biased (Beznosikov et al., 2020).

Table 1: UpCom complexity ($\alpha = 0$) of linearly converging algorithms with LT or CC and allowing for PP (with exact gradients). The $\widetilde{\mathcal{O}}$ notation hides the $\log \epsilon^{-1}$ factor (and other log factors for Scaffold). $c \in \{2, \ldots, n\}$ is the number of participating clients and the other notations are recalled in Table 3.

| Algorithm | LT | CC | UpCom |
|---|---|---|---|
| DIANA-PP [a] | ✗ | ✓ | $\widetilde{\mathcal{O}}\big((1 + \frac{d}{c})\kappa + d\frac{n}{c}\big)$ |
| Scaffold | ✓ | ✗ | $\widetilde{\mathcal{O}}(d\kappa + d\frac{n}{c})$ |
| 5GCS | ✓ | ✗ | $\widetilde{\mathcal{O}}\big(d\sqrt{\kappa}\sqrt{\frac{n}{c}} + d\frac{n}{c}\big)$ |
| TAMUNA | ✓ | ✓ | $\widetilde{\mathcal{O}}\big(\sqrt{d}\sqrt{\kappa}\sqrt{\frac{n}{c}} + d\sqrt{\kappa}\frac{\sqrt{n}}{c} + d\frac{n}{c}\big)$ |

[a] using independent `rand`-1 compressors, for instance. Note that $\mathcal{O}(\sqrt{d}\sqrt{\kappa}\sqrt{\frac{n}{c}} + d\frac{n}{c})$ is better than $\mathcal{O}(\kappa + d\frac{n}{c})$ and $\mathcal{O}(d\sqrt{\kappa}\frac{\sqrt{n}}{c} + d\frac{n}{c})$ is better than $\mathcal{O}(\frac{d}{c}\kappa + d\frac{n}{c})$, so that TAMUNA has a better complexity than DIANA-PP.

Table 2: TotalCom complexity of linearly converging algorithms using Local Training (LT), Communication Compression (CC), or both, in case of full participation and exact gradients. The $\widetilde{\mathcal{O}}$ notation hides the $\log \epsilon^{-1}$ factor. The notations are recalled in Table 3.

| Algorithm | LT | CC | TotalCom | TotalCom=UpCom when $\alpha = 0$ |
|---|---|---|---|---|
| DIANA [a] | ✗ | ✓ | $\widetilde{\mathcal{O}}\big((1 + \alpha d + \frac{d + \alpha d^2}{n})\kappa + d + \alpha d^2\big)$ | $\widetilde{\mathcal{O}}\big((1 + \frac{d}{n})\kappa + d\big)$ |
| EF21 [b] | ✗ | ✓ | $\widetilde{\mathcal{O}}(d\kappa)$ | $\widetilde{\mathcal{O}}(d\kappa)$ |
| Scaffold | ✓ | ✗ | $\widetilde{\mathcal{O}}(d\kappa)$ | $\widetilde{\mathcal{O}}(d\kappa)$ |
| FedLin | ✓ | ✗ | $\widetilde{\mathcal{O}}(d\kappa)$ | $\widetilde{\mathcal{O}}(d\kappa)$ |
| S-Local-GD | ✓ | ✗ | $\widetilde{\mathcal{O}}(d\kappa)$ | $\widetilde{\mathcal{O}}(d\kappa)$ |
| Scaffnew | ✓ | ✗ | $\widetilde{\mathcal{O}}(d\sqrt{\kappa})$ | $\widetilde{\mathcal{O}}(d\sqrt{\kappa})$ |
| 5GCS | ✓ | ✗ | $\widetilde{\mathcal{O}}(d\sqrt{\kappa})$ | $\widetilde{\mathcal{O}}(d\sqrt{\kappa})$ |
| FedCOMGATE | ✓ | ✓ | $\widetilde{\mathcal{O}}(d\kappa)$ | $\widetilde{\mathcal{O}}(d\kappa)$ |
| TAMUNA | ✓ | ✓ | $\widetilde{\mathcal{O}}\big(\sqrt{d}\sqrt{\kappa} + d\frac{\sqrt{\kappa}}{\sqrt{n}} + d + \sqrt{\alpha}\,d\sqrt{\kappa}\big)$ | $\widetilde{\mathcal{O}}\big(\sqrt{d}\sqrt{\kappa} + d\frac{\sqrt{\kappa}}{\sqrt{n}} + d\big)$ |

[a] using independent `rand`-1 compressors, for instance. Note that $\mathcal{O}(\sqrt{d}\sqrt{\kappa} + d)$ is better than $\mathcal{O}(\kappa + d)$ and $\mathcal{O}(d\frac{\sqrt{\kappa}}{\sqrt{n}} + d)$ is better than $\mathcal{O}(\frac{d}{n}\kappa + d)$, so that TAMUNA has a better complexity than DIANA.

[b] using `top`-$k$ compressors with any $k$, for instance.

The variance-reduced algorithm DIANA (Mishchenko et al., 2019) is a major contribution to the field, as it converges linearly with a large class of unbiased compressors. For instance, when the clients use independent `rand`-1 compressors for UpCom, the UpCom complexity of DIANA is $\mathcal{O}\big((\kappa(1 + \frac{d}{n}) + d)\log \epsilon^{-1}\big)$. If $n$ is large, this is much better than with GD. DIANA was later extended in several ways (Horváth et al., 2022; Gorbunov et al., 2020a); in particular, DIANA-PP is a generalized version allowing for PP (Condat & Richtárik, 2022). Algorithms converging linearly with biased compressors have been proposed recently, like EF21 (Richtárik et al., 2021; Fatkhullin et al., 2021; Condat et al., 2022b), but the theory is less mature and the acceleration potential not as clear as with unbiased compressors. We summarize existing results in Table 2. Our algorithm TAMUNA benefits from CC with specific unbiased compressors, with even more acceleration than DIANA. Also, the focus in DIANA is on UpCom and its DownCom step is the same as in GD, with the full model broadcast at every iteration, so that its TotalCom complexity can be *worse* than the one of GD. Extensions of DIANA with bidirectional CC, i.e. compression in both UpCom and DownCom, have been proposed (Gorbunov et al., 2020b; Philippenko & Dieuleveut, 2020; Liu et al., 2020; Condat & Richtárik, 2022), but this does not improve its TotalCom complexity; see also Philippenko & Dieuleveut (2021) and references therein on bidirectional CC. We note that if LT is disabled ($\mathcal{L}^{(r)} \equiv 1$), TAMUNA is still new and does not revert to a known algorithm with CC.

---

**Algorithm 1** TAMUNA

1: **input:** stepsizes $\gamma > 0$, $\eta > 0$; number of participating clients $c \in \{2, \ldots, n\}$; sparsity index for compression $s \in \{2, \ldots, c\}$; initial model estimate $\bar{x}^{(0)} \in \mathbb{R}^d$ at the server and initial control variates $h_1^{(0)}, \ldots, h_n^{(0)} \in \mathbb{R}^d$ at the clients, such that $\sum_{i=1}^n h_i^{(0)} = 0$.

2: **for** $r = 0, 1, \ldots$ (rounds) **do**

3:   choose a subset $\Omega^{(r)} \subset [n]$ of size $c$ uniformly at random

4:   choose the number of local steps $\mathcal{L}^{(r)} \geq 1$

5:   **for** clients $i \in \Omega^{(r)}$, in parallel, **do**

6:     $x_i^{(r,0)} := \bar{x}^{(r)}$ (initialization received from the server)

7:     **for** $\ell = 0, \ldots, \mathcal{L}^{(r)} - 1$ (local steps) **do**

8:       $x_i^{(r,\ell+1)} := x_i^{(r,\ell)} - \gamma g_i^{(r,\ell)} + \gamma h_i^{(r)}$, where $g_i^{(r,\ell)}$ is an unbiased stochastic estimate of $\nabla f_i\big(x_i^{(r,\ell)}\big)$ of variance $\sigma_i^2$

9:     **end for**

10:   **end for**

11:   UpCom: the server and active clients agree on a random binary mask $\mathbf{q}^{(r)} = \big(q_i^{(r)}\big)_{i \in \Omega^{(r)}} \in \mathbb{R}^{d \times c}$ generated as explained in Figure 1, and every client $i \in \Omega^{(r)}$ sends the compressed vector $\mathcal{C}_i^{(r)}\Big(x_i^{(r,\mathcal{L}^{(r)})}\Big)$ to the server, where $\mathcal{C}_i^{(r)}(v)$ denotes $v$ multiplied elementwise by $q_i^{(r)}$.

12:   $\bar{x}^{(r+1)} := \frac{1}{s} \sum_{i \in \Omega^{(r)}} \mathcal{C}_i^{(r)}\Big(x_i^{(r,\mathcal{L}^{(r)})}\Big)$ (aggregation by the server)

13:   **for** clients $i \in \Omega^{(r)}$, in parallel, **do**

14:     $h_i^{(r+1)} := h_i^{(r)} + \frac{\eta}{\gamma}\Big(\mathcal{C}_i^{(r)}\big(\bar{x}^{(r+1)}\big) - \mathcal{C}_i^{(r)}\big(x_i^{(r,\mathcal{L}^{(r)})}\big)\Big)$ ($\bar{x}^{(r+1)}$ is received from the server)

15:   **end for**

16:   **for** clients $i \notin \Omega^{(r)}$, in parallel, **do**

17:     $h_i^{(r+1)} := h_i^{(r)}$ (the client is idle)

18:   **end for**

19: **end for**

---

## 1.4 Challenges and contributions

Our new algorithm TAMUNA builds upon the LT mechanism of Scaffnew and enables PP and CC, which are essential features for applicability to real-world FL setups. In short,

$$\text{TAMUNA} = \underbrace{\text{(S)GD} + \text{LT}}_{\text{Scaffnew}} + \text{PP} + \text{CC}.$$

We focus on the strongly convex setting but we also prove sublinear convergence of TAMUNA in the merely convex case in the Appendix. We emphasize that the problem can be arbitrarily heterogeneous: we do not make any assumption on the functions $f_i$ beyond smoothness and strong convexity, and there is no notion of data similarity whatsoever. We also stress that our goal is to deepen our theoretical understanding of LT, CC and PP, and to make these 3 intuitive and effective mechanisms, which are widely used in practice, work in the best possible way when harnessed to (stochastic) GD. Thus, we establish convergence of TAMUNA in Theorem 1 with stochastic GD steps of bounded variance, which is more general than exact GD steps, but only to illustrate the robustness of our framework. A thorough analysis would need to consider the general setting of possibly variance-reduced (Gorbunov et al., 2020a; Gower et al., 2020) SGD local steps, as was done for Scaffnew in Malinovsky et al. (2022). We leave it for future work, since we focus on the *communication* complexity, and stochastic gradients can only *worsen* it. Reducing the *computation* complexity using accelerated (Nesterov, 2004) or stochastic GD steps is somewhat orthogonal to our present study.

Let us elaborate on the double challenge of combining LT with PP and CC. Our notations are summarized in Table 3 for convenience.

Table 3: Summary of the main notations used in the paper.

| | |
|---|---|
| LT | local training |
| CC | communication compression |
| PP | partial participation (a.k.a. client sampling) |
| $L$ | smoothness constant |
| $\mu$ | strong convexity constant |
| $\kappa = L/\mu$ | condition number of the functions |
| $d$ | dimension of the model |
| $n, i$ | number and index of clients |
| $[n] = \{1, \ldots, n\}$ | |
| $\alpha$ | weight on downlink communication (DownCom), see (2) |
| $\sigma_i^2, \sigma^2 := \sum_i \sigma_i^2$ | variance of the stochastic gradients, see (3) |
| $c \in \{2, \ldots, n\}$ | number of active clients (a.k.a. cohort size). Full participation if $c = n$ |
| $\Omega \subset [n]$ | index set of active clients |
| $s \in \{2, \ldots, c\}$ | sparsity index for compression. No compression if $s = c$ |
| $\mathbf{q} = (q_i)_{i=1}^c$ | random binary mask for compression, as detailed in Figure 1 |
| $r$ | index of rounds |
| $\mathcal{L}, \ell$ | number and index of local steps in a round |
| $p$ | inverse of the expected number of local steps per round |
| $t, T$ | indexes of iterations |
| $\gamma, \eta, \chi$ | stepsizes |
| $x_i$ | local model estimate at client $i$ |
| $h_i$ | local control variate tracking $\nabla f_i$ |
| $\bar{x}^{(r)}$ | model estimate at the server at round $r$ |
| $\tau$ | convergence rate |

### 1.4.1 Combining LT and PP

With the recent breakthrough of Scaffnew (Mishchenko et al., 2022), we now understand that LT is not only efficient in practice, but also grounded in theory, and yields communication acceleration if the number of local steps is chosen appropriately. However, Scaffnew does not allow for PP. It has been an open and challenging question to know whether its powerful randomized mechanism would be compatible with PP. In fact, according to Grudzień et al. (2023), the authors of Scaffnew *"have tried—very hard in their own words— but their efforts did not bear any fruit."* In this paper, we break this lock: TAMUNA handles LT and PP, and fully benefits from the acceleration of LT, whatever the participation level; that is, its communication complexity depends on $\sqrt{\kappa}$, not $\kappa$.

Combining LT and PP is difficult: we want PP not only during communication whenever it occurs, but also with respect to all computations before. The simple idea of allowing at every round some clients to be active and to proceed normally, and other clients to be idle with unchanged local variables, does not work. A key property of TAMUNA is that only the clients which participated in a given round make use of the updated model broadcast by the server to update their control variates (step 14). From a mathematical point of view, our approach relies on combining the two stochastic processes of probabilistic communication and random client selection *in two different ways*, for updating after communication the model estimates $x_i$ on one hand, and the control variates $h_i$ on the other hand. Indeed, a crucial property is that the sum of the control variates over all clients always remains zero. This separate treatment was the key to the success of our design.

### 1.4.2 Combining LT and CC

It is very challenging to combine LT and CC. In the strongly convex and heterogeneous case considered here, the methods Qsparse-local-SGD (Basu et al., 2020) and FedPAQ (Reisizadeh et al., 2020) do not converge linearly. The only linearly converging LT + CC algorithm we are aware of is FedCOMGATE (Haddadpour et al., 2021). But its rate is $\mathcal{O}(d\kappa \log \epsilon^{-1})$, which does not show any acceleration. We note that random

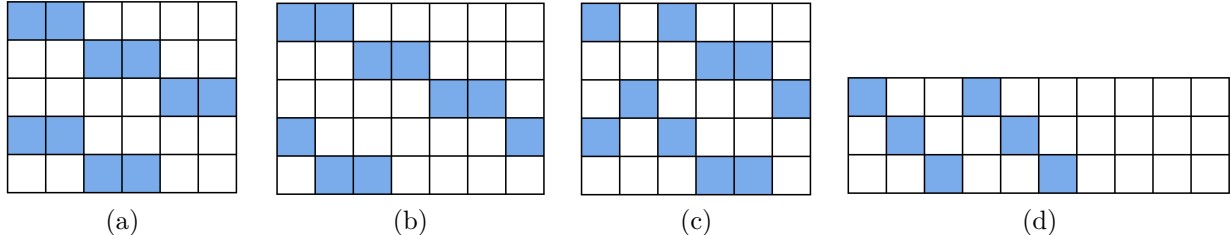

Figure 1: The random sampling pattern $\mathbf{q}^{(r)} = (q_i^{(r)})_{i=1}^c \in \mathbb{R}^{d \times c}$ used for communication is generated by a random permutation of the columns of a fixed binary template pattern, which has the prescribed number $s \geq 2$ of ones in every row. In (a) with $(d, c, s) = (5, 6, 2)$ and (b) with $(d, c, s) = (5, 7, 2)$, with ones in blue and zeros in white, examples of the template pattern used when $d \geq \frac{c}{s}$: for every row $k \in [d]$, there are $s$ ones at columns $i = \mod(s(k-1), c) + 1, \ldots, \mod(sk - 1, c) + 1$. Thus, there are $\lfloor \frac{sd}{c} \rfloor$ or $\lceil \frac{sd}{c} \rceil$ ones in every column vector $q_i$. In (c), an example of sampling pattern obtained after a permutation of the columns of the template pattern in (a). In (d) with $(d, c, s) = (3, 10, 2)$, an example of the template pattern used when $\frac{c}{s} \geq d$: for every column $i = 1, \ldots, ds$, there is 1 one at row $k = \mod(i-1, d) + 1$. Thus, there is 0 or 1 one in every column vector $q_i$. We can note that when $d = \frac{c}{s}$, the two different rules for $d \geq \frac{c}{s}$ and $\frac{c}{s} \geq d$ for constructing the template pattern are equivalent, since they give exactly the same set of sampling patterns when permuting their columns. These two rules make it possible to generate easily the columns $q_i^{(r)}$ of $\mathbf{q}^{(r)}$ on the fly, without having to generate the whole mask $\mathbf{q}^{(r)}$ explicitly.

reshuffling, which can be seen as a kind of LT, has been combined with CC in Sadiev et al. (2022b); Malinovsky & Richtárik (2022).

Like for PP, the program of combining LT and CC looks simple, as it seems we just have to "plug" compressors into Scaffnew. Again, this simple approach does not work and the key is to have separate stochastic mechanisms to update the local model estimates and the control variates. In Condat et al. (2022a), a specific compression mechanism compatible with LT was designed and CompressedScaffnew, which combines the LT mechanism of Scaffnew and this new CC mechanism, was proposed. CompressedScaffnew is the first algorithm, to the best of our knowledge, to exhibit a doubly-accelerated linear rate, by leveraging LT and CC. However, like Scaffnew, CompressedScaffnew only works in case of full participation. We stress that this successful combination of LT and CC does not help in combining LT and PP: a non-participating client does not participate to communication whenever it occurs, but it also does not perform any computation before. Therefore, there is no way to enable PP in loopless algorithms like Scaffnew and CompressedScaffnew, where communication can be triggered at any time. Whether a client participates or not must be decided in advance, at the beginning of a round consisting of a sequence of local steps followed by communication. Our new algorithm TAMUNA is the first to solve this challenge. It works with any level of PP, with as few as two clients participating in every round. TAMUNA relies on the same dedicated design of the compressors as CompressedScaffnew, explained in Figure 1 and such that the messages sent by the different clients complement each other, to keep a tight control of the variance after aggregation. We currently do not know how to use any other type of compressors in TAMUNA.

Thus, by combining LT and CC, TAMUNA establishes the new state of the art in communication efficiency. For instance, with exact gradients, if $\alpha$ is small and $n$ is large, its TotalCom complexity in case of full participation is

$$\mathcal{O}\left( \left( \sqrt{d}\sqrt{\kappa} + d \right) \log \epsilon^{-1} \right);$$

our general result is in Theorem 2. Thus, TAMUNA enjoys twofold acceleration, with $\sqrt{\kappa}$ instead of $\kappa$ thanks to LT and $\sqrt{d}$ instead of $d$ thanks to CC.

## 2   Proposed algorithm TAMUNA

The proposed algorithm TAMUNA is shown as Algorithm 1. Its main loop is over the rounds, indexed by $r$. A round consists of a sequence, written as an inner loop, of local steps indexed by $\ell$ and performed in parallel by the active clients, followed by compressed communication with the server and update of the local control variates $h_i$. The $c$ active, or participating, clients are selected randomly at the beginning of the round. During UpCom, every client sends a compressed version of its local model $x_i$: it sends only a few of its elements, selected randomly according to the rule explained in Figure 1 and known by both the clients and the server (for decoding).

At the end of the round, the aggregated model estimate $\bar{x}^{(r+1)}$ formed by the server is sent only to the active clients, which use it to update their control variates $h_i$. This update consists in overwriting only the coordinates of $h_i$ which have been involved in the communication process; that is, for which the mask $q_i^{(r)}$ has a one. Indeed, the received vector $\bar{x}^{(r+1)}$ does not contain relevant information to update $h_i$ at the other coordinates.

The update of the local model estimates $x_i$ at the clients takes place at the beginning of the round, when the active clients download the current model estimate $\bar{x}^{(r)}$ to initialize their local steps. So, it seems that there are two DownCom steps from the server to the clients per round (steps 6 and 14), but the algorithm can be written with only one: $\bar{x}^{(r+1)}$ can be broadcast by the server at the end of round $r$ not only to the active clients of round $r$, but also to the active clients of the next round $r+1$, at the same time. We keep the algorithm written in this way for simplicity.

Thus, the clients of index $i \notin \Omega^{(r)}$, which do not participate in round $r$, are completely idle: they do not compute and do not communicate at all. Their local control variates $h_i$ remain unchanged, and they do not even need to store a local model estimate: they only need to receive the latest model estimate $x^{(r)}$ from the server when they participate in the process.

In TAMUNA, unbiased stochastic gradient estimates of bounded variance $\sigma_i^2$ can be used: for every $i \in [n]$,

$$\mathbb{E}\Big[g_i^{(r,\ell)} \mid x_i^{(r,\ell)}\Big] = \nabla f_i\big(x_i^{(r,\ell)}\big), \quad \mathbb{E}\Big[\big\|g_i^{(r,\ell)} - \nabla f_i\big(x_i^{(r,\ell)}\big)\big\|^2 \mid x_i^{(r,\ell)}\Big] \le \sigma_i^2, \tag{3}$$

for some $\sigma_i \ge 0$. We have $g_i^{(r,\ell)} = \nabla f_i\big(x_i^{(r,\ell)}\big)$ if $\sigma_i = 0$. We define the total variance $\sigma^2 := \sum_{i=1}^{n} \sigma_i^2$. Our main result, stating linear convergence of TAMUNA to the exact solution $x^\star$ of (1), or to a neighborhood if $\sigma > 0$, is the following:

**Theorem 1** (fast linear convergence to a $\sigma^2$-neighborhood). *Let $p \in (0, 1]$. In TAMUNA, suppose that at every round $r \ge 0$, $\mathcal{L}^{(r)}$ is chosen randomly and independently according to a geometric law of mean $p^{-1}$; that is, for every $\mathcal{L} \ge 1$, $\mathrm{Prob}(\mathcal{L}^{(r)} = \mathcal{L}) = (1-p)^{\mathcal{L}-1}p$. Also, suppose that*

$$0 < \gamma < \frac{2}{L} \tag{4}$$

*and $\eta := p\chi$, where*

$$0 < \chi \le \frac{n(s-1)}{s(n-1)} \in \left(\frac{1}{2}, 1\right]. \tag{5}$$

*For every total number $t \ge 0$ of local steps made so far, define the Lyapunov function*

$$\overline{\Psi}^t := \frac{n}{\gamma}\big\|\bar{x}^t - x^\star\big\|^2 + \frac{\gamma}{p^2\chi}\frac{n-1}{s-1}\sum_{i=1}^{n}\big\|h_i^{(r)} - h_i^\star\big\|^2, \tag{6}$$

*where $x^\star$ is the unique solution to (1), $h_i^\star = \nabla f_i(x^\star)$, $r \ge 0$ and $\ell \in \{0, \dots, \mathcal{L}^{(r)} - 1\}$ are such that*

$$t = \sum_{\hat{r}=0}^{r-1} \mathcal{L}^{(\hat{r})} + \ell, \tag{7}$$

*and*

$$\bar{x}^t := \frac{1}{s} \sum_{i \in \Omega^{(r)}} \mathcal{C}_i^{(r)} \left( x_i^{(r,\ell)} \right). \tag{8}$$

*Then, for every $t \geq 0$,*

$$\mathbb{E}\left[\overline{\Psi}^t\right] \leq \tau^t \overline{\Psi}^0 + \frac{\gamma \sigma^2}{1 - \tau}, \tag{9}$$

*where*

$$\tau := \max\left( (1 - \gamma\mu)^2, (\gamma L - 1)^2, 1 - p^2 \chi \frac{s-1}{n-1} \right) < 1. \tag{10}$$

*Also, if $\sigma = 0$, $(\bar{x}^{(r)})_{r \in \mathbb{N}}$ converges to $x^\star$ and $(h_i^{(r)})_{r \in \mathbb{N}}$ converges to $h_i^\star$, almost surely.*

The complete proof is in the Appendix. We give a brief sketch here. The analysis is made for a single-loop version of the algorithm, shown as Algorithm 2, with a loop over the iterations, indexed by $t$, and one local step per iteration. Thus, communication does not happen at every iteration but is only triggered randomly with probability $p$. Its convergence is proved in Theorem 3. Indeed, the contraction of the Lyapunov function happens at every iteration and not at every round, whose size is random. That is why we have to reindex the local steps to obtain a rate depending on the number of iterations $t$ so far. We detail in the Appendix how Theorem 3 on Algorithm 2 yields Theorem 1 on TAMUNA.

We note that in (8), $\bar{x}^t$ is actually computed only if $\ell = 0$, in which case $\bar{x}^t = \bar{x}^{(r)}$. We also note that the theorem depends on $s$ but not on $c$. The dependence on $c$ is hidden in the fact that $s$ is upper bounded by $c$.

**Remark 1** (setting $\eta$). *In the conditions of Theorem 1, one can simply set $\eta = \frac{p}{2}$ in TAMUNA, which is independent of $n$ and $s$. However, the larger $\eta$, the better, so it is recommended to set*

$$\eta = p \frac{n(s-1)}{s(n-1)}. \tag{11}$$

*Also, as a rule of thumb, if the average number of local steps per round is $\mathcal{L}$, one can replace $p$ by $\mathcal{L}^{-1}$.*

We can comment on the difference between TAMUNA and Scaffold, when CC is disabled ($s = c$). In TAMUNA, $h_i$ is updated by adding $\bar{x}^{(r+1)} - x_i^{(r,\mathcal{L}^{(r)})}$, the difference between the latest global estimate $\bar{x}^{(r+1)}$ and the latest local estimate $x_i^{(r,\mathcal{L}^{(r)})}$. By contrast, in Scaffold, $\bar{x}^{(r)} - x_i^{(r,\mathcal{L}^{(r)})}$ is used instead, which involves the "old" global estimate $\bar{x}^{(r)}$. Moreover, this difference is scaled by the number of local steps, which makes it small. That is why no acceleration from LT can be obtained in Scaffold, whatever the number of local steps. This is not a weakness of the analysis in Karimireddy et al. (2020) but an intrinsic limitation of Scaffold.

We can also note that the neighborhood size in (9) does not show so-called linear speedup; that is, it does not decrease when $n$ increases. The properties of performing LT with SGD steps remain little understood (Woodworth et al., 2020), and we believe this should be studied within the general framework of variance reduction (Malinovsky et al., 2022). Again, this goes far beyond the scope of this paper, which focuses on communication.

## 3 Iteration and communication complexities

We consider in this section that exact gradients are used ($\sigma = 0$),[2] since our aim is to establish a new state of the art for the communication complexity, regardless of the type of local computations. We place ourselves in the conditions of Theorem 1.

We first remark that TAMUNA has the same iteration complexity as GD, with rate $\tau^\sharp := \max(1 - \gamma\mu, \gamma L - 1)^2$, as long as $p$ and $s$ are large enough to have $1 - \chi p^2 \frac{s-1}{n-1} \leq \tau^\sharp$. This is remarkable: LT, CC and PP do not harm convergence at all, until some threshold.

---

[2]If $\sigma > 0$, it is possible to derive sublinear rates to reach $\epsilon$-accuracy for the communication complexity, by setting $\gamma$ proportional to $\epsilon$, as was done for Scaffnew in Mishchenko et al. (2022, Corollary 5.6).

Let us consider the number of iterations (= total number of local steps) to reach $\epsilon$-accuracy, i.e. $\mathbb{E}\left[\overline{\Psi}^t\right] \leq \epsilon$. For any $s \geq 2$, $p \in (0, 1]$, $\gamma = \Theta(\frac{1}{L})$, and $\chi = \Theta(1)$, the iteration complexity of TAMUNA is

$$\mathcal{O}\left(\left(\kappa + \frac{n}{sp^2}\right)\log \epsilon^{-1}\right).$$

Thus, by choosing

$$p = \min\left(\Theta\left(\sqrt{\frac{n}{s\kappa}}\right), 1\right), \tag{12}$$

which means that the average number of local steps per round is

$$\mathbb{E}\left[\mathcal{L}^{(r)}\right] = \max\left(\Theta\left(\sqrt{\frac{s\kappa}{n}}\right), 1\right), \tag{13}$$

the iteration complexity becomes

$$\mathcal{O}\left(\left(\kappa + \frac{n}{s}\right)\log \epsilon^{-1}\right).$$

We now consider the communication complexity. Communication occurs at every iteration with probability $p$, and during every communication round, DownCom consists in broadcasting the full $d$-dimensional vector $\bar{x}^{(r)}$, whereas in UpCom, compression is effective and the number of real values sent in parallel by the clients is equal to the number of ones per column in the sampling pattern $\mathbf{q}$, which is $\lceil \frac{sd}{c} \rceil \geq 1$. Hence, the communication complexities are:

$$\text{DownCom:} \quad \mathcal{O}\left(pd\left(\kappa + \frac{n}{sp^2}\right)\log \epsilon^{-1}\right),$$

$$\text{UpCom:} \quad \mathcal{O}\left(p\left(\frac{sd}{c} + 1\right)\left(\kappa + \frac{n}{sp^2}\right)\log \epsilon^{-1}\right).$$

$$\text{TotalCom:} \quad \mathcal{O}\left(p\left(\frac{sd}{c} + 1 + \alpha d\right)\left(\kappa + \frac{n}{sp^2}\right)\log \epsilon^{-1}\right).$$

For a given $s$, the best choice for $p$, for both DownCom and UpCom, is given in (12), for which

$$\mathcal{O}\left(p\left(\kappa + \frac{n}{sp^2}\right)\right) = \mathcal{O}\left(\sqrt{\frac{n\kappa}{s}} + \frac{n}{s}\right)$$

and the TotalCom complexity is

$$\text{TotalCom:} \quad \mathcal{O}\left(\left(\sqrt{\frac{n\kappa}{s}} + \frac{n}{s}\right)\left(\frac{sd}{c} + 1 + \alpha d\right)\log \epsilon^{-1}\right).$$

We see the first acceleration effect due to LT: with a suitable $p < 1$, the communication complexity only depends on $\sqrt{\kappa}$, not $\kappa$, whatever the participation level $c$ and compression level $s$.

Without compression, i.e. $s = c$, whatever $\alpha$, the TotalCom complexity becomes

$$\mathcal{O}\left(d\left(\sqrt{\frac{n\kappa}{c}} + \frac{n}{c}\right)\log \epsilon^{-1}\right).$$

We can now set $s$ to further accelerate the algorithm, by minimizing the TotalCom complexity:

**Theorem 2** (doubly accelerated communication)**.** *In the conditions of Theorem 1, suppose that $\sigma = 0$, $\gamma = \Theta(\frac{1}{L})$, $\chi = \Theta(1)$, and*

$$p = \min\left(\Theta\left(\sqrt{\frac{n}{s\kappa}}\right), 1\right), \quad s = \max\left(2, \left\lfloor\frac{c}{d}\right\rfloor, \lfloor \alpha c \rfloor\right). \tag{14}$$

*Then the TotalCom complexity of* TAMUNA *is*

$$\mathcal{O}\left(\left(\sqrt{d}\sqrt{\kappa}\sqrt{\frac{n}{c}} + d\sqrt{\kappa}\frac{\sqrt{n}}{c} + d\frac{n}{c} + \sqrt{\alpha}\,d\sqrt{\kappa}\sqrt{\frac{n}{c}}\right)\log \epsilon^{-1}\right). \tag{15}$$

As reported in Tables 1 and 2, TAMUNA improves upon all known algorithms using either LT or CC on top of GD, even those working only with full participation.

**Corollary 1** (dependence on $\alpha$). *As long as $\alpha \leq \max(\frac{2}{c}, \frac{1}{d}, \frac{n}{\kappa c})$, there is no difference with the case $\alpha = 0$, in which we only focus on UpCom, and the TotalCom complexity is*

$$\mathcal{O}\left(\left(\sqrt{d}\sqrt{\kappa}\sqrt{\frac{n}{c}} + d\sqrt{\kappa}\frac{\sqrt{n}}{c} + d\frac{n}{c}\right)\log \epsilon^{-1}\right). \tag{16}$$

*On the other hand, if $\alpha \geq \max(\frac{2}{c}, \frac{1}{d}, \frac{n}{\kappa c})$, the complexity increases and becomes*

$$\mathcal{O}\left(\sqrt{\alpha}d\sqrt{\kappa}\sqrt{\frac{n}{c}}\log \epsilon^{-1}\right), \tag{17}$$

*but compression remains operational and effective with the $\sqrt{\alpha}$ factor. It is only when $\alpha = 1$ that $s = c$, i.e. there is no compression, and that the Upcom, DownCom and TotalCom complexities all become*

$$\mathcal{O}\left(d\sqrt{\kappa}\sqrt{\frac{n}{c}}\log \epsilon^{-1}\right). \tag{18}$$

*Thus, in case of full participation ($c = n$), TAMUNA is faster than Scaffnew for every $\alpha \in [0, 1]$.*

**Corollary 2** (full participation). *In case of full participation ($c = n$), the TotalCom complexity of TAMUNA is*

$$\mathcal{O}\left(\left(\sqrt{d}\sqrt{\kappa} + d\frac{\sqrt{\kappa}}{\sqrt{n}} + d + \sqrt{\alpha}\,d\sqrt{\kappa}\right)\log \epsilon^{-1}\right). \tag{19}$$

## 4 Experiments

Our work is theoretical and studies the foundational properties of a class of algorithms. Nonetheless, we carry experiments to illustrate our results using a practical logistic regression problem.

The global loss function is defined as

$$f(x) = \frac{1}{M}\sum_{m=1}^{M}\left(\log\big(1 + \exp\big(-b_m a_m^\top x\big)\big) + \frac{\mu}{2}\|x\|^2\right), \tag{20}$$

where the variables $a_m \in \mathbb{R}^d$ and $b_m \in \{-1, 1\}$ represent the data samples, and $M$ denotes the total number of samples. The function $f$ in (20) is divided into $n$ separate functions $f_i$, with any remainder from dividing $M$ by $n$ discarded.

We select the strong convexity constant $\mu$ so that $\kappa = 10^4$.

For our analysis, we choose $n = 1000$ and examine two scenarios: in the first one, we have $d > n$ using the 'real-sim' dataset with $d = 20958$, and in the second one, we have $n > d$ using the 'w8a' dataset with $d = 300$, from the widely-used LIBSVM library (Chang & Lin, 2011). Additionally, we consider two cases for each scenario: $\alpha = 0$ and $\alpha = 0.1$, where $\alpha$ is the weight on DownCom defined in (2).

We measure the convergence error $f(x) - f(x^\star)$ with respect to TotalCom, i.e. the total number of communicated reals, as defined in Section (1.2). Here, $x$ denotes the model known by the server; for TAMUNA, this is $\bar{x}^{(r)}$. This error serves as a natural basis for comparing algorithms, and since $f$ is $L$-smooth, we have $f(x) - f(x^\star) \leq \frac{L}{2}\|x - x^\star\|^2$ for any $x$. Consequently, the error converges linearly at the same rate as $\Psi$ in Theorem 1..

We compare the performance of three algorithms allowing for PP, namely Scaffold, 5GCS, and TAMUNA, for two participation scenarios: $c = n$ and $c = 0.1n$ (10% participation). In the full participation case, we add Scaffnew to the comparison.

In order to ensure theoretical conditions that guarantee linear convergence, we set $\gamma$ and $\eta$ for TAMUNA as

$$\gamma = \frac{2}{L + \mu}, \quad \eta = p\frac{n(s-1)}{s(n-1)},$$

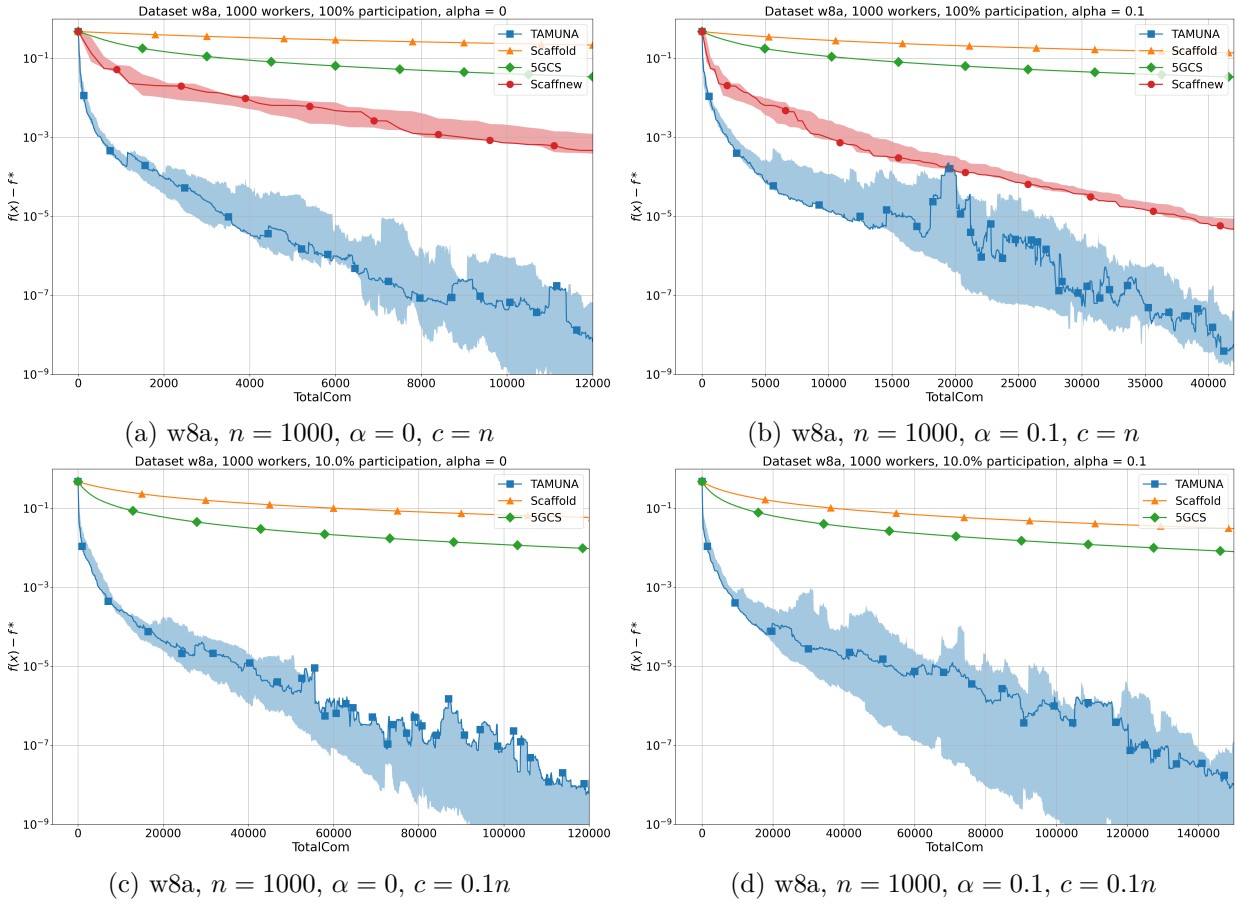

Figure 2: Logistic regression experiment in the case $n > d$. The dataset w8a has $d = 300$ features and $n = 1000$, so $n \approx 3d$. The first row shows a comparison in the full participation regime, while the second row shows a comparison in the partial participation regime with 10% of clients. On the left, $\alpha = 0$, while on the right, $\alpha = 0.1$.

where the remaining parameters $s$ and $p$ are fine-tuned to achieve the best communication complexity. In our experimental setup, we found that using $s = 40$ and $p = 0.01$ resulted in excellent performance. The conditions of Theorem 1 are met with these values, so linear convergence of TAMUNA is guaranteed. We adopt the same values of $\gamma$ and $p$ for Scaffnew. For Scaffold, we use $p^{-1}$ local steps, which is the same, on average, as for TAMUNA and Scaffnew; the behavior of Scaffold changed marginally with other values. We also set $\gamma$ to its highest value that ensures convergence. In the case of 5GCS, we tune $\gamma$, $\tau$, and the number of local steps to achieve the best communication complexity.

The models in all algorithms, as well as the control variates in TAMUNA, Scaffnew and Scaffold, are initialized with zero vectors.

The results are shown in Figures 2 and 3. Each algorithm is run multiple times with different random seeds, depending on its variance (7 times for TAMUNA, 5 times for Scaffnew, and 3 times for Scaffold and 5GCS). The shaded area in the plots shows the difference between the maximum and minimum convergence error achieved over these runs. Additionally, the progress of the first run for each algorithm is depicted with a thicker line and markers.

As can be seen, our proposed algorithm TAMUNA outperforms all other methods. In case of full participation, Scaffnew outperforms Scaffold and 5GCS, which shows the efficiency of its LT mechanism. TAMUNA embeds the same mechanism and also benefits from it, but it outperforms Scaffnew thanks to CC, its second

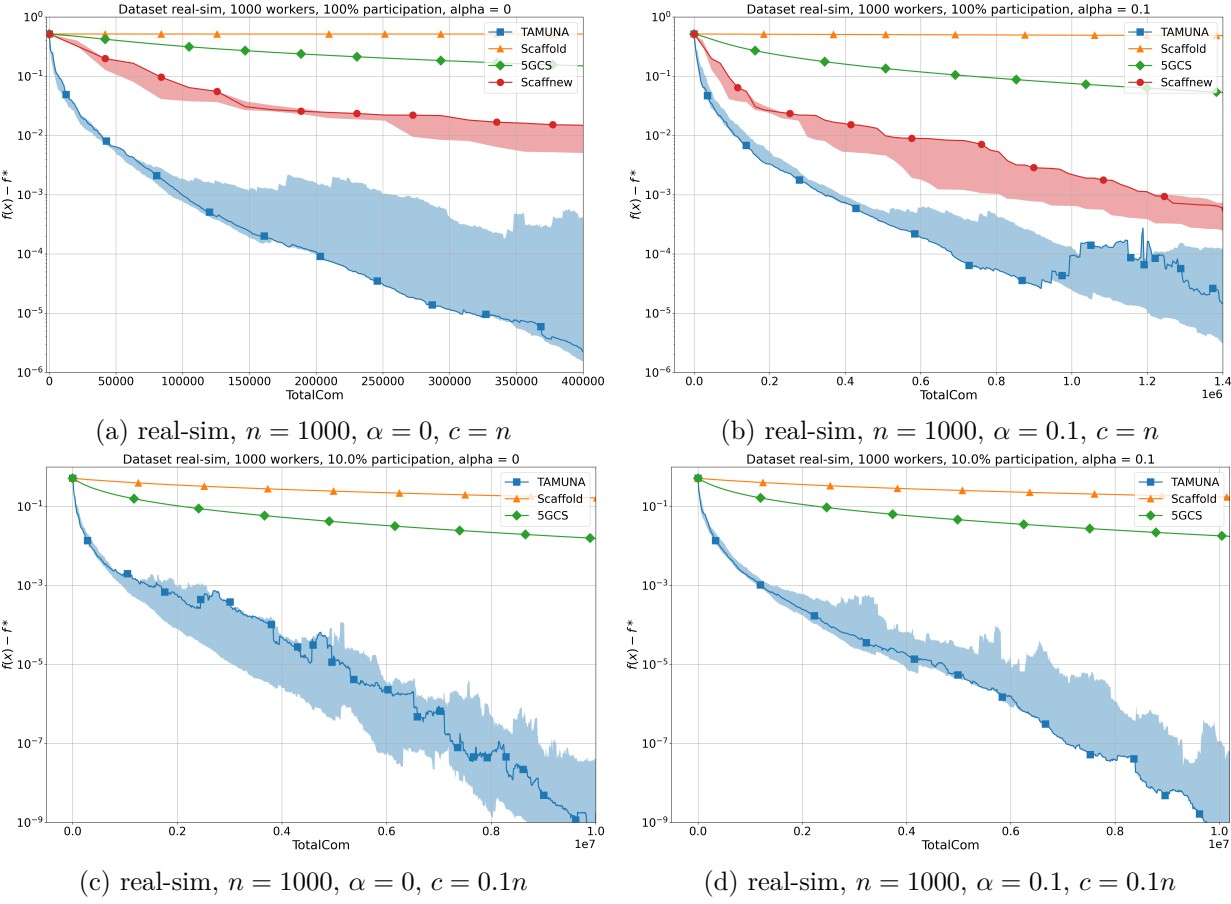

Figure 3: Logistic regression experiment in the case $d > n$. The dataset real-sim has $d = 20,958$ features and $n = 1000$, so $n \approx d/20$. The first row shows a comparison in the full participation regime, while the second row shows a comparison in the partial participation regime with $10\%$ of clients. On the left, $\alpha = 0$, while on the right, $\alpha = 0.1$.

communication-acceleration mechanism. The difference between TAMUNA and Scaffnew is larger for $\alpha = 0$ than for $\alpha = 0.1$, as explained by our theory; the difference would vanish if $\alpha$ tends to 1. TAMUNA is applicable and proved to converge with any level of PP, whereas Scaffnew only applies to the full participation case.

## 5    Conclusion

We have proposed TAMUNA, the first communication-efficient algorithm that allows for partial participation (PP) and provably benefits from the two combined acceleration mechanisms of Local Training (LT) and Communication Compression (CC), in the convex setting. Moreover, this is achieved not only for uplink communication, but for our more comprehensive model of total communication. These theoretical guarantees are confirmed in practice and TAMUNA communicates less than existing algorithms to reach the same accuracy. An important venue for future work will be to generalize our specific compression mechanism to a broad class of compressors including quantization (Horváth et al., 2022). Another venue consists in implementing internal variance reduction for the stochastic gradients, as was done for Scaffnew in Malinovsky et al. (2022). Analyzing the properties of TAMUNA on nonconvex problems should also be studied (Karimireddy et al., 2021; Das et al., 2022).

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

---

**Algorithm 2**

---

1: **input:** stepsizes $\gamma > 0$, $\chi > 0$; probability $p \in (0,1]$; number of participating clients $c \in \{2,\ldots,n\}$; compression index $s \in \{2,\ldots,c\}$; initial estimates $x_1^0,\ldots,x_n^0 \in \mathbb{R}^d$ and $h_1^0,\ldots,h_n^0 \in \mathbb{R}^d$ such that $\sum_{i=1}^n h_i^0 = 0$, sequence of independent coin flips $\theta^0, \theta^1,\ldots$ with $\mathrm{Prob}(\theta^t = 1) = p$, and for every $t$ with $\theta^t = 1$, a subset $\Omega^t \subset [n]$ of size $c$ chosen uniformly at random and a random binary mask $\mathbf{q}^t = (q_i^t)_{i \in \Omega^t} \in \mathbb{R}^{d \times c}$ generated as explained in Figure 1. The compressed vector $\mathcal{C}_i^t(v)$ is $v$ multiplied elementwise by $q_i^t$.

2: **for** $t = 0, 1, \ldots$ **do**

3:    **for** $i = 1,\ldots,n$, at clients in parallel, **do**

4:       $\hat{x}_i^t := x_i^t - \gamma g_i^t + \gamma h_i^t$, where $g_i^t$ is an unbiased stochastic estimate of $\nabla f_i(x_i^t)$ of variance $\sigma_i^2$

5:       **if** $\theta^t = 1$ **then**

6:          **if** $i \in \Omega^t$ **then**

7:             send $\hat{x}_i^t$ to the server, which aggregates $\bar{x}^t := \frac{1}{s}\sum_{j \in \Omega^t} \mathcal{C}_j^t(\hat{x}_j^t)$ and broadcasts it to all clients

8:             $h_i^{t+1} := h_i^t + \frac{p\chi}{\gamma}\big(\mathcal{C}_i^t(\bar{x}^t) - \mathcal{C}_i^t(\hat{x}_i^t)\big)$

9:          **else**

10:             $h_i^{t+1} := h_i^t$

11:          **end if**

12:          $x_i^{t+1} := \bar{x}^t$

13:       **else**

14:          $x_i^{t+1} := \hat{x}_i^t$

15:          $h_i^{t+1} := h_i^t$

16:       **end if**

17:    **end for**

18: **end for**

---

# Appendix

## A   Proof of Theorem 1

We first prove convergence of Algorithm 2, which is a single-loop version of TAMUNA; that is, there is a unique loop over the iterations and there is one local step per iteration. In Section A.2, we show that this yields a proof of Theorem 1 for TAMUNA. We can note that in case of full participation ($c = n$, $\Omega^t \equiv [n]$), Algorithm 2 reverts to CompressedScaffnew (Condat et al., 2022a).

To simplify the analysis of Algorithm 2, we introduce vector notations: the problem (1) can be written as

$$\text{find } \mathbf{x}^\star = \underset{\mathbf{x} \in \mathcal{X}}{\arg\min} \ \mathbf{f}(\mathbf{x}) \quad \text{s.t.} \quad W\mathbf{x} = 0, \tag{21}$$

where $\mathcal{X} := \mathbb{R}^{d \times n}$, an element $\mathbf{x} = (x_i)_{i=1}^n \in \mathcal{X}$ is a collection of vectors $x_i \in \mathbb{R}^d$, $\mathbf{f}: \mathbf{x} \in \mathcal{X} \mapsto \sum_{i=1}^n f_i(x_i)$ is $L$-smooth and $\mu$-strongly convex, the linear operator $W: \mathcal{X} \to \mathcal{X}$ maps $\mathbf{x} = (x_i)_{i=1}^n$ to $(x_i - \frac{1}{n}\sum_{j=1}^n x_j)_{i=1}^n$. The constraint $W\mathbf{x} = 0$ means that $\mathbf{x}$ minus its average is zero; that is, $\mathbf{x}$ has identical components $x_1 = \cdots = x_n$. Thus, (21) is indeed equivalent to (1). We have $W = W^* = W^2$.

We also rewrite Algorithm 2 using vector notations as Algorithm 3. It converges linearly:

**Theorem 3** (fast linear convergence). *In Algorithm 3, suppose that $0 < \gamma < \frac{2}{L}$, $0 < \chi \leq \frac{n(s-1)}{s(n-1)}$, $\omega = \frac{n-1}{p(s-1)} - 1$. For every $t \geq 0$, define the Lyapunov function*

$$\Psi^t := \frac{1}{\gamma}\left\|\mathbf{x}^t - \mathbf{x}^\star\right\|^2 + \frac{\gamma(1+\omega)}{p\chi}\left\|\mathbf{h}^t - \mathbf{h}^\star\right\|^2, \tag{22}$$

*where $\mathbf{x}^\star$ is the unique solution to (21) and $\mathbf{h}^\star := \nabla \mathbf{f}(\mathbf{x}^\star)$. Then Algorithm 3 converges linearly: for every $t \geq 0$,*

$$\mathbb{E}\big[\Psi^t\big] \leq \tau^t \Psi^0 + \frac{\gamma \sigma^2}{1 - \tau}, \tag{23}$$

---

**Algorithm 3**

**input:** stepsizes $\gamma > 0$, $\chi > 0$; probability $p \in (0,1]$, parameter $\omega \geq 0$; number of participating clients $c \in \{2,\ldots,n\}$; compression index $s \in \{2,\ldots,c\}$; initial estimates $\mathbf{x}^0 \in \mathcal{X}$ and $\mathbf{h}^0 \in \mathcal{X}$ such that $\sum_{i=1}^n h_i^0 = 0$; sequence of independent coin flips $\theta^0, \theta^1, \ldots$ with $\mathrm{Prob}(\theta^t = 1) = p$, and for every $t$ with $\theta^t = 1$, a subset $\Omega^t \subset [n]$ of size $c$ chosen uniformly at random and a random binary mask $\mathbf{q}^t = (q_i^t)_{i \in \Omega^t} \in \mathbb{R}^{d \times c}$ generated as explained in Figure 1. The compressed vector $\mathcal{C}_i^t(v)$ is $v$ multiplied elementwise by $q_i^t$.

**for** $t = 0, 1, \ldots$ **do**

    $\hat{\mathbf{x}}^t := \mathbf{x}^t - \gamma \mathbf{g}^t + \gamma \mathbf{h}^t$, where $\mathbf{g}^t = (g_i^t)_{i=1}^n \approx \nabla \mathbf{f}(\mathbf{x}^t)$

    **if** $\theta^t = 1$ **then**

        $\bar{\mathbf{x}}^t := (\bar{x}^t)_{i=1}^n$, where $\bar{x}^t := \frac{1}{s} \sum_{j \in \Omega^t} \mathcal{C}_j^t(\hat{x}_j^t)$

        $\mathbf{x}^{t+1} := \bar{\mathbf{x}}^t$

        $\mathbf{d}^t := (d_i^t)_{i=1}^n$ with $d_i^t = \begin{cases} (1+\omega)\left(\mathcal{C}_i^t(\hat{x}_i^t) - \mathcal{C}_i^t(\bar{x}^t)\right) & \text{if } i \in \Omega^t, \\ 0 & \text{otherwise} \end{cases}$

    **else**

        $\mathbf{x}^{t+1} := \hat{\mathbf{x}}^t$

        $\mathbf{d}^t := 0$

    **end if**

    $\mathbf{h}^{t+1} := \mathbf{h}^t - \frac{p\chi}{\gamma(1+\omega)}\mathbf{d}^t$

**end for**

---

*where*

$$\tau := \max\left((1-\gamma\mu)^2, (\gamma L - 1)^2, 1 - p^2\chi\frac{s-1}{n-1}\right) < 1. \tag{24}$$

*Also, if $\sigma = 0$, $(\mathbf{x}^t)_{t \in \mathbb{N}}$ and $(\hat{\mathbf{x}}^t)_{t \in \mathbb{N}}$ both converge to $\mathbf{x}^\star$ and $(\mathbf{h}^t)_{t \in \mathbb{N}}$ converges to $\mathbf{h}^\star$, almost surely.*

*Proof.* We consider the variables of Algorithm 3. For every $t \geq 0$, we denote by $\mathcal{F}_0^t$ the $\sigma$-algebra generated by the collection of $\mathcal{X}$-valued random variables $\mathbf{x}^0, \mathbf{h}^0, \ldots, \mathbf{x}^t, \mathbf{h}^t$, and by $\mathcal{F}^t$ the $\sigma$-algebra generated by these variables, as well as the stochastic gradients $\mathbf{g}^t$. $\mathbf{d}^t$ is a random variable; as proved in Section A.1, it satisfies the 3 following properties, on which the convergence analysis of Algorithm 3 relies: for every $t \geq 0$,

1. $\mathbb{E}[\mathbf{d}^t \mid \mathcal{F}^t] = W\hat{\mathbf{x}}^t$.

2. $\mathbb{E}\left[\left\|\mathbf{d}^t - W\hat{\mathbf{x}}^t\right\|^2 \mid \mathcal{F}^t\right] \leq \omega\left\|W\hat{\mathbf{x}}^t\right\|^2$.

3. $\mathbf{d}^t$ belongs to the range of $W$; that is, $\sum_{i=1}^n d_i^t = 0$.

We suppose that $\sum_{i=1}^n h_i^0 = 0$. Then, it follows from the third property of $\mathbf{d}^t$ that, for every $t \geq 0$, $\sum_{i=1}^n h_i^t = 0$; that is, $W\mathbf{h}^t = \mathbf{h}^t$.

For every $t \geq 0$, we define $\hat{\mathbf{h}}^{t+1} := \mathbf{h}^t - \frac{p\chi}{\gamma}W\hat{\mathbf{x}}^t$, $\mathbf{w}^t := \mathbf{x}^t - \gamma\mathbf{g}^t$ and $\mathbf{w}^\star := \mathbf{x}^\star - \gamma\nabla\mathbf{f}(\mathbf{x}^\star)$. We also define $\bar{\mathbf{x}}^{t\sharp} := (\bar{x}^{t\sharp})_{i=1}^n$, with $\bar{x}^{t\sharp} := \frac{1}{n}\sum_{i=1}^n \hat{x}_i^t$; that is, $\bar{x}^{t\sharp}$ is the exact average of the $\hat{x}_i^t$, of which $\bar{x}^t$ is an unbiased random estimate.

Let $t \geq 0$. We have

$$\mathbb{E}\left[\left\|\mathbf{x}^{t+1} - \mathbf{x}^\star\right\|^2 \mid \mathcal{F}^t\right] = p\mathbb{E}\left[\left\|\bar{\mathbf{x}}^t - \mathbf{x}^\star\right\|^2 \mid \mathcal{F}^t, \theta^t = 1\right] + (1-p)\left\|\hat{\mathbf{x}}^t - \mathbf{x}^\star\right\|^2,$$

Since $\mathbb{E}[\bar{\mathbf{x}}^t \mid \mathcal{F}^t, \theta^t = 1] = \bar{\mathbf{x}}^{t\sharp}$,

$$\mathbb{E}\left[\left\|\bar{\mathbf{x}}^t - \mathbf{x}^\star\right\|^2 \mid \mathcal{F}^t, \theta^t = 1\right] = \left\|\bar{\mathbf{x}}^{t\sharp} - \mathbf{x}^\star\right\|^2 + \mathbb{E}\left[\left\|\bar{\mathbf{x}}^t - \bar{\mathbf{x}}^{t\sharp}\right\|^2 \mid \mathcal{F}^t, \theta = 1\right],$$

with

$$\left\|\bar{\mathbf{x}}^{t\sharp} - \mathbf{x}^\star\right\|^2 = \left\|\hat{\mathbf{x}}^t - \mathbf{x}^\star\right\|^2 - \left\|W\hat{\mathbf{x}}^t\right\|^2.$$

To analyze $\mathbb{E}\left[\left\|\bar{\mathbf{x}}^t - \mathbf{x}^\star\right\|^2 \mid \mathcal{F}^t, \theta^t = 1\right]$, where the expectation is with respect to the active subset $\Omega^t$ and the mask $\mathbf{q}^t$, we can remark that the expectation and the squared Euclidean norm are separable with respect to the coordinates of the $d$-dimensional vectors. So, we can reason on the coordinates independently on each other, even if the the coordinates, or rows, of $\mathbf{q}^t$ are mutually dependent. Also, for a given coordinate $k \in [d]$, choosing $s$ elements at random among the $c$ elements $\hat{x}_{i,k}^t$ with $i \in \Omega^t$, with $\Omega^t$ chosen uniformly at random too, is equivalent to selecting $s$ elements $\hat{x}_{i,k}^t$ among all $i \in [n]$ uniformly at random in the first place. Thus, for every coordinate $k \in [d]$, it is like a subset $\widetilde{\Omega}_k^t \subset [n]$ of size $s$, which corresponds to the location of the ones in the $k$-th row of $\mathbf{q}^t$, is chosen uniformly at random and

$$\bar{x}_k^t = \frac{1}{s} \sum_{i \in \widetilde{\Omega}_k^t} \hat{x}_{i,k}^t.$$

Then, as proved in Condat & Richtárik (2022, Proposition 1),

$$\mathbb{E}\left[\left\|\bar{\mathbf{x}}^t - \bar{\mathbf{x}}^{t\sharp}\right\|^2 \mid \mathcal{F}^t, \theta^t = 1\right] = n \sum_{k=1}^d \mathbb{E}_{\widetilde{\Omega}_k^t}\left[\left(\frac{1}{s} \sum_{i \in \widetilde{\Omega}_k^t} \hat{x}_{i,k}^t - \frac{1}{n} \sum_{j=1}^n \hat{x}_{j,k}^t\right)^2 \mid \mathcal{F}^t\right] = \nu \left\|W\hat{\mathbf{x}}^t\right\|^2,$$

where

$$\nu := \frac{n-s}{s(n-1)} \in \left[0, \frac{1}{2}\right). \tag{25}$$

Moreover,

$$\begin{aligned}
\left\|\hat{\mathbf{x}}^t - \mathbf{x}^\star\right\|^2 &= \left\|\mathbf{w}^t - \mathbf{w}^\star\right\|^2 + \gamma^2 \left\|\mathbf{h}^t - \mathbf{h}^\star\right\|^2 + 2\gamma\langle\mathbf{w}^t - \mathbf{w}^\star, \mathbf{h}^t - \mathbf{h}^\star\rangle \\
&= \left\|\mathbf{w}^t - \mathbf{w}^\star\right\|^2 - \gamma^2 \left\|\mathbf{h}^t - \mathbf{h}^\star\right\|^2 + 2\gamma\langle\hat{\mathbf{x}}^t - \mathbf{x}^\star, \mathbf{h}^t - \mathbf{h}^\star\rangle \\
&= \left\|\mathbf{w}^t - \mathbf{w}^\star\right\|^2 - \gamma^2 \left\|\mathbf{h}^t - \mathbf{h}^\star\right\|^2 + 2\gamma\langle\hat{\mathbf{x}}^t - \mathbf{x}^\star, \hat{\mathbf{h}}^{t+1} - \mathbf{h}^\star\rangle - 2\gamma\langle\hat{\mathbf{x}}^t - \mathbf{x}^\star, \hat{\mathbf{h}}^{t+1} - \mathbf{h}^t\rangle \\
&= \left\|\mathbf{w}^t - \mathbf{w}^\star\right\|^2 - \gamma^2 \left\|\mathbf{h}^t - \mathbf{h}^\star\right\|^2 + 2\gamma\langle\hat{\mathbf{x}}^t - \mathbf{x}^\star, \hat{\mathbf{h}}^{t+1} - \mathbf{h}^\star\rangle + 2p\chi\langle\hat{\mathbf{x}}^t - \mathbf{x}^\star, W\hat{\mathbf{x}}^t\rangle \\
&= \left\|\mathbf{w}^t - \mathbf{w}^\star\right\|^2 - \gamma^2 \left\|\mathbf{h}^t - \mathbf{h}^\star\right\|^2 + 2\gamma\langle\hat{\mathbf{x}}^t - \mathbf{x}^\star, \hat{\mathbf{h}}^{t+1} - \mathbf{h}^\star\rangle + 2p\chi\left\|W\hat{\mathbf{x}}^t\right\|^2.
\end{aligned}$$

Hence,

$$\begin{aligned}
\mathbb{E}\left[\left\|\mathbf{x}^{t+1} - \mathbf{x}^\star\right\|^2 \mid \mathcal{F}^t\right] &= p\left\|\hat{\mathbf{x}}^t - \mathbf{x}^\star\right\|^2 - p\left\|W\hat{\mathbf{x}}^t\right\|^2 + p\nu\left\|W\hat{\mathbf{x}}^t\right\|^2 + (1-p)\left\|\hat{\mathbf{x}}^t - \mathbf{x}^\star\right\|^2 \\
&= \left\|\hat{\mathbf{x}}^t - \mathbf{x}^\star\right\|^2 - p(1-\nu)\left\|W\hat{\mathbf{x}}^t\right\|^2 \\
&= \left\|\mathbf{w}^t - \mathbf{w}^\star\right\|^2 - \gamma^2\left\|\mathbf{h}^t - \mathbf{h}^\star\right\|^2 + 2\gamma\langle\hat{\mathbf{x}}^t - \mathbf{x}^\star, \hat{\mathbf{h}}^{t+1} - \mathbf{h}^\star\rangle \\
&\quad + \left(2p\chi - p(1-\nu)\right)\left\|W\hat{\mathbf{x}}^t\right\|^2.
\end{aligned}$$

On the other hand, using the 3 properties of $\mathbf{d}^t$ stated above, we have

$$
\begin{aligned}
\mathbb{E}\left[\left\|\mathbf{h}^{t+1} - \mathbf{h}^\star\right\|^2 \mid \mathcal{F}^t\right] &\leq \left\|\mathbf{h}^t - \mathbf{h}^\star - \frac{p\chi}{\gamma(1+\omega)} W\hat{\mathbf{x}}^t\right\|^2 + \frac{\omega p^2 \chi^2}{\gamma^2(1+\omega)^2}\left\|W\hat{\mathbf{x}}^t\right\|^2 \\
&= \left\|\mathbf{h}^t - \mathbf{h}^\star + \frac{1}{1+\omega}(\hat{\mathbf{h}}^{t+1} - \mathbf{h}^t)\right\|^2 + \frac{\omega}{(1+\omega)^2}\left\|\hat{\mathbf{h}}^{t+1} - \mathbf{h}^t\right\|^2 \\
&= \left\|\frac{\omega}{1+\omega}(\mathbf{h}^t - \mathbf{h}^\star) + \frac{1}{1+\omega}(\hat{\mathbf{h}}^{t+1} - \mathbf{h}^\star)\right\|^2 + \frac{\omega}{(1+\omega)^2}\left\|\hat{\mathbf{h}}^{t+1} - \mathbf{h}^t\right\|^2 \\
&= \frac{\omega^2}{(1+\omega)^2}\left\|\mathbf{h}^t - \mathbf{h}^\star\right\|^2 + \frac{1}{(1+\omega)^2}\left\|\hat{\mathbf{h}}^{t+1} - \mathbf{h}^\star\right\|^2 \\
&\quad + \frac{2\omega}{(1+\omega)^2}\langle\mathbf{h}^t - \mathbf{h}^\star, \hat{\mathbf{h}}^{t+1} - \mathbf{h}^\star\rangle + \frac{\omega}{(1+\omega)^2}\left\|\hat{\mathbf{h}}^{t+1} - \mathbf{h}^\star\right\|^2 \\
&\quad + \frac{\omega}{(1+\omega)^2}\left\|\mathbf{h}^t - \mathbf{h}^\star\right\|^2 - \frac{2\omega}{(1+\omega)^2}\langle\mathbf{h}^t - \mathbf{h}^\star, \hat{\mathbf{h}}^{t+1} - \mathbf{h}^\star\rangle \\
&= \frac{1}{1+\omega}\left\|\hat{\mathbf{h}}^{t+1} - \mathbf{h}^\star\right\|^2 + \frac{\omega}{1+\omega}\left\|\mathbf{h}^t - \mathbf{h}^\star\right\|^2 .
\end{aligned}
$$

Moreover,

$$
\begin{aligned}
\left\|\hat{\mathbf{h}}^{t+1} - \mathbf{h}^\star\right\|^2 &= \left\|(\mathbf{h}^t - \mathbf{h}^\star) + (\hat{\mathbf{h}}^{t+1} - \mathbf{h}^t)\right\|^2 \\
&= \left\|\mathbf{h}^t - \mathbf{h}^\star\right\|^2 + \left\|\hat{\mathbf{h}}^{t+1} - \mathbf{h}^t\right\|^2 + 2\langle\mathbf{h}^t - \mathbf{h}^\star, \hat{\mathbf{h}}^{t+1} - \mathbf{h}^t\rangle \\
&= \left\|\mathbf{h}^t - \mathbf{h}^\star\right\|^2 + 2\langle\hat{\mathbf{h}}^{t+1} - \mathbf{h}^\star, \hat{\mathbf{h}}^{t+1} - \mathbf{h}^t\rangle - \left\|\hat{\mathbf{h}}^{t+1} - \mathbf{h}^t\right\|^2 \\
&= \left\|\mathbf{h}^t - \mathbf{h}^\star\right\|^2 - \left\|\hat{\mathbf{h}}^{t+1} - \mathbf{h}^t\right\|^2 - 2\frac{p\chi}{\gamma}\langle\hat{\mathbf{h}}^{t+1} - \mathbf{h}^\star, W(\hat{\mathbf{x}}^t - \mathbf{x}^\star)\rangle \\
&= \left\|\mathbf{h}^t - \mathbf{h}^\star\right\|^2 - \frac{p^2\chi^2}{\gamma^2}\left\|W\hat{\mathbf{x}}^t\right\|^2 - 2\frac{p\chi}{\gamma}\langle W(\hat{\mathbf{h}}^{t+1} - \mathbf{h}^\star), \hat{\mathbf{x}}^t - \mathbf{x}^\star\rangle \\
&= \left\|\mathbf{h}^t - \mathbf{h}^\star\right\|^2 - \frac{p^2\chi^2}{\gamma^2}\left\|W\hat{\mathbf{x}}^t\right\|^2 - 2\frac{p\chi}{\gamma}\langle\hat{\mathbf{h}}^{t+1} - \mathbf{h}^\star, \hat{\mathbf{x}}^t - \mathbf{x}^\star\rangle .
\end{aligned}
$$

Hence,

$$
\begin{aligned}
\frac{1}{\gamma}\mathbb{E}\left[\left\|\mathbf{x}^{t+1} - \mathbf{x}^\star\right\|^2 \mid \mathcal{F}^t\right] &+ \frac{\gamma(1+\omega)}{p\chi}\mathbb{E}\left[\left\|\mathbf{h}^{t+1} - \mathbf{h}^\star\right\|^2 \mid \mathcal{F}^t\right] \\
&\leq \frac{1}{\gamma}\left\|\mathbf{w}^t - \mathbf{w}^\star\right\|^2 - \gamma\left\|\mathbf{h}^t - \mathbf{h}^\star\right\|^2 + \left(2\frac{p\chi}{\gamma} - \frac{p}{\gamma}(1-\nu)\right)\left\|W\hat{\mathbf{x}}^t\right\|^2 \\
&\quad + 2\langle\hat{\mathbf{x}}^t - \mathbf{x}^\star, \hat{\mathbf{h}}^{t+1} - \mathbf{h}^\star\rangle + \frac{\gamma}{p\chi}\left\|\mathbf{h}^t - \mathbf{h}^\star\right\|^2 \\
&\quad - \frac{p\chi}{\gamma}\left\|W\hat{\mathbf{x}}^t\right\|^2 - 2\langle\hat{\mathbf{h}}^{t+1} - \mathbf{h}^\star, \hat{\mathbf{x}}^t - \mathbf{x}^\star\rangle + \frac{\gamma\omega}{p\chi}\left\|\mathbf{h}^t - \mathbf{h}^\star\right\|^2 \\
&= \frac{1}{\gamma}\left\|\mathbf{w}^t - \mathbf{w}^\star\right\|^2 + \left(\frac{\gamma(1+\omega)}{p\chi} - \gamma\right)\left\|\mathbf{h}^t - \mathbf{h}^\star\right\|^2 \\
&\quad + \left(\frac{p\chi}{\gamma} - \frac{p(1-\nu)}{\gamma}\right)\left\|W\hat{\mathbf{x}}^t\right\|^2 . \quad\quad (26)
\end{aligned}
$$

Since we have supposed

$$
0 < \chi \leq 1 - \nu = \frac{n(s-1)}{s(n-1)} \in \left(\frac{1}{2}, 1\right],
$$

we have

$$\frac{1}{\gamma}\mathbb{E}\Big[\big\|\mathbf{x}^{t+1} - \mathbf{x}^\star\big\|^2 \mid \mathcal{F}^t\Big] + \frac{\gamma(1+\omega)}{p\chi}\mathbb{E}\Big[\big\|\mathbf{h}^{t+1} - \mathbf{h}^\star\big\|^2 \mid \mathcal{F}^t\Big]$$

$$\leq \frac{1}{\gamma}\big\|\mathbf{w}^t - \mathbf{w}^\star\big\|^2 + \frac{\gamma(1+\omega)}{p\chi}\left(1 - \frac{p\chi}{1+\omega}\right)\big\|\mathbf{h}^t - \mathbf{h}^\star\big\|^2.$$

Finally,

$$\mathbb{E}\Big[\big\|\mathbf{w}^t - \mathbf{w}^\star\big\|^2 \mid \mathcal{F}_0^t\Big] \leq \big\|(\mathrm{Id} - \gamma\nabla\mathbf{f})\mathbf{x}^t - (\mathrm{Id} - \gamma\nabla\mathbf{f})\mathbf{x}^\star\big\|^2 + \gamma^2\sigma^2,$$

and according to Condat & Richtárik (2023, Lemma 1),

$$\big\|(\mathrm{Id} - \gamma\nabla\mathbf{f})\mathbf{x}^t - (\mathrm{Id} - \gamma\nabla\mathbf{f})\mathbf{x}^\star\big\|^2 \leq \max(1 - \gamma\mu, \gamma L - 1)^2 \big\|\mathbf{x}^t - \mathbf{x}^\star\big\|^2.$$

Therefore,

$$\mathbb{E}\big[\Psi^{t+1} \mid \mathcal{F}_0^t\big] \leq \max\left((1-\gamma\mu)^2, (\gamma L - 1)^2, 1 - \frac{p\chi}{1+\omega}\right)\Psi^t + \gamma\sigma^2$$

$$= \max\left((1-\gamma\mu)^2, (\gamma L - 1)^2, 1 - p^2\chi\frac{s-1}{n-1}\right)\Psi^t + \gamma\sigma^2. \tag{27}$$

Using the tower rule, we can unroll the recursion in (27) to obtain the unconditional expectation of $\Psi^{t+1}$.

If $\sigma = 0$, using classical results on supermartingale convergence (Bertsekas, 2015, Proposition A.4.5), it follows from (27) that $\Psi^t \to 0$ almost surely. Almost sure convergence of $\mathbf{x}^t$ and $\mathbf{h}^t$ follows. Finally, by Lipschitz continuity of $\nabla\mathbf{f}$, we can upper bound $\|\hat{\mathbf{x}}^t - \mathbf{x}^\star\|^2$ by a linear combination of $\|\mathbf{x}^t - \mathbf{x}^\star\|^2$ and $\|\mathbf{h}^t - \mathbf{h}^\star\|^2$. It follows that $\mathbb{E}\Big[\|\hat{\mathbf{x}}^t - \mathbf{x}^\star\|^2\Big] \to 0$ linearly with the same rate $\tau$ and that $\hat{\mathbf{x}}^t \to \mathbf{x}^\star$ almost surely, as well. $\qquad\square$

## A.1 The random variable $\mathbf{d}^t$

We study the random variable $\mathbf{d}^t$ used in Algorithm 3. If $\theta^t = 0$, $\mathbf{d}^t = 0$. If, on the other hand, $\theta^t = 1$, for every coordinate $k \in [d]$, a subset $\widetilde{\Omega}_k^t \subset [n]$ of size $s$ is chosen uniformly at random. These sets $(\widetilde{\Omega}_k^t)_{k=1}^d$ are mutually dependent, but this does not matter for the derivations, since we can reason on the coordinates separately. Then, for every $k \in [d]$ and $i \in [n]$,

$$d_{i,k}^t \coloneqq \begin{cases} a\left(\hat{x}_{i,k}^t - \frac{1}{s}\sum_{j\in\widetilde{\Omega}_k^t}\hat{x}_{j,k}^t\right) & \text{if } i \in \widetilde{\Omega}_k^t, \\ 0 & \text{otherwise}, \end{cases} \tag{28}$$

for some value $a > 0$ to determine. We can check that $\sum_{i=1}^n d_i^t = 0$. We can also note that $\mathbf{d}^t$ depends only on $W\hat{\mathbf{x}}^t$ and not on $\hat{\mathbf{x}}^t$; in particular, if $\hat{x}_1^t = \cdots = \hat{x}_n^t$, $d_i^t = 0$. We have to set $a$ so that $\mathbb{E}[d_i^t] = \hat{x}_i^t - \frac{1}{n}\sum_{j=1}^n \hat{x}_j^t$, where the expectation is with respect to $\theta^t$ and the $\widetilde{\Omega}_k^t$ (all expectations in this section are conditional to $\hat{\mathbf{x}}^t$). So, let us calculate this expectation.

Let $k \in [d]$. For every $i \in [n]$,

$$\mathbb{E}\big[d_{i,k}^t\big] = p\frac{s}{n}\left(a\hat{x}_{i,k}^t - \frac{a}{s}\mathbb{E}_{\Omega:i\in\Omega}\left[\sum_{j\in\Omega}\hat{x}_{j,k}^t\right]\right),$$

where $\mathbb{E}_{\Omega:i\in\Omega}$ denotes the expectation with respect to a subset $\Omega \subset [n]$ of size $s$ containing $i$ and chosen uniformly at random. We have

$$\mathbb{E}_{\Omega:i\in\Omega}\left[\sum_{j\in\Omega}\hat{x}_{j,k}^t\right] = \hat{x}_{i,k}^t + \frac{s-1}{n-1}\sum_{j\in[n]\setminus\{i\}}\hat{x}_{j,k}^t = \frac{n-s}{n-1}\hat{x}_{i,k}^t + \frac{s-1}{n-1}\sum_{j=1}^n\hat{x}_{j,k}^t.$$

Hence, for every $i \in [n]$,

$$\mathbb{E}\left[d_{i,k}^t\right] = p\frac{s}{n}\left(a - \frac{a}{s}\frac{n-s}{n-1}\right)\hat{x}_{i,k} - p\frac{s}{n}\frac{a}{s}\frac{s-1}{n-1}\sum_{j=1}^n \hat{x}_{j,k}.$$

Therefore, by setting

$$a := \frac{n-1}{p(s-1)}, \tag{29}$$

we have, for every $i \in [n]$,

$$\mathbb{E}\left[d_{i,k}^t\right] = p\frac{s}{n}\left(\frac{1}{p}\frac{n-1}{s-1} - \frac{1}{p}\frac{n-s}{s(s-1)}\right)\hat{x}_{i,k} - \frac{1}{n}\sum_{j=1}^n \hat{x}_{j,k}$$

$$= \hat{x}_{i,k} - \frac{1}{n}\sum_{j=1}^n \hat{x}_{j,k},$$

as desired.

Now, we want to find the value of $\omega$ such that

$$\mathbb{E}\left[\left\|\mathbf{d}^t - W\hat{\mathbf{x}}^t\right\|^2\right] \leq \omega\left\|W\hat{\mathbf{x}}^t\right\|^2 \tag{30}$$

or, equivalently,

$$\mathbb{E}\left[\sum_{i=1}^n \left\|d_i^t\right\|^2\right] \leq (1+\omega)\sum_{i=1}^n \left\|\hat{x}_i^t - \frac{1}{n}\sum_{j=1}^n \hat{x}_j^t\right\|^2.$$

We can reason on the coordinates separately, or all at once to ease the notations. We have

$$\mathbb{E}\left[\sum_{i=1}^n \left\|d_i^t\right\|^2\right] = p\frac{s}{n}\sum_{i=1}^n \mathbb{E}_{\Omega:i\in\Omega}\left\|a\hat{x}_i^t - \frac{a}{s}\sum_{j\in\Omega}\hat{x}_j^t\right\|^2.$$

For every $i \in [n]$,

$$\mathbb{E}_{\Omega:i\in\Omega}\left\|a\hat{x}_i^t - \frac{a}{s}\sum_{j\in\Omega}\hat{x}_j^t\right\|^2 = \mathbb{E}_{\Omega:i\in\Omega}\left\|\left(a-\frac{a}{s}\right)\hat{x}_i^t - \frac{a}{s}\sum_{j\in\Omega\setminus\{i\}}\hat{x}_j^t\right\|^2$$

$$= \left\|\left(a-\frac{a}{s}\right)\hat{x}_i^t\right\|^2 + \mathbb{E}_{\Omega:i\in\Omega}\left\|\frac{a}{s}\sum_{j\in\Omega\setminus\{i\}}\hat{x}_j^t\right\|^2$$

$$- 2\left\langle\left(a-\frac{a}{s}\right)\hat{x}_i^t, \frac{a}{s}\mathbb{E}_{\Omega:i\in\Omega}\sum_{j\in\Omega\setminus\{i\}}\hat{x}_j^t\right\rangle.$$

We have

$$\mathbb{E}_{\Omega:i\in\Omega}\sum_{j\in\Omega\setminus\{i\}}\hat{x}_j^t = \frac{s-1}{n-1}\sum_{j\in[n]\setminus\{i\}}\hat{x}_j^t = \frac{s-1}{n-1}\left(\sum_{j=1}^n \hat{x}_j^t - \hat{x}_i^t\right)$$

and

$$
\begin{aligned}
\mathbb{E}_{\Omega:i\in\Omega}\left\|\sum_{j\in\Omega\setminus\{i\}}\hat{x}_j^t\right\|^2 &= \mathbb{E}_{\Omega:i\in\Omega}\sum_{j\in\Omega\setminus\{i\}}\left\|\hat{x}_j^t\right\|^2 + \mathbb{E}_{\Omega:i\in\Omega}\sum_{j\in\Omega\setminus\{i\}}\sum_{j'\in\Omega\setminus\{i,j\}}\langle\hat{x}_j^t,\hat{x}_{j'}^t\rangle \\
&= \frac{s-1}{n-1}\sum_{j\in[n]\setminus\{i\}}\left\|\hat{x}_j^t\right\|^2 + \frac{s-1}{n-1}\frac{s-2}{n-2}\sum_{j\in[n]\setminus\{i\}}\sum_{j'\in[n]\setminus\{i,j\}}\langle\hat{x}_j^t,\hat{x}_{j'}^t\rangle \\
&= \frac{s-1}{n-1}\left(1-\frac{s-2}{n-2}\right)\sum_{j\in[n]\setminus\{i\}}\left\|\hat{x}_j^t\right\|^2 + \frac{s-1}{n-1}\frac{s-2}{n-2}\left\|\sum_{j\in[n]\setminus\{i\}}\hat{x}_j^t\right\|^2 \\
&= \frac{s-1}{n-1}\frac{n-s}{n-2}\left(\sum_{j=1}^n\left\|\hat{x}_j^t\right\|^2 - \left\|\hat{x}_i^t\right\|^2\right) + \frac{s-1}{n-1}\frac{s-2}{n-2}\left\|\sum_{j=1}^n\hat{x}_j^t - \hat{x}_i^t\right\|^2 .
\end{aligned}
$$

Hence,

$$
\begin{aligned}
\mathbb{E}\left[\sum_{i=1}^n\left\|d_i^t\right\|^2\right] &= p\frac{s}{n}\sum_{i=1}^n\left\|\left(a-\frac{a}{s}\right)\hat{x}_i^t\right\|^2 + ps\frac{a^2}{(s)^2}\frac{s-1}{n-1}\frac{n-s}{n-2}\sum_{j=1}^n\left\|\hat{x}_j^t\right\|^2 \\
&\quad - p\frac{s}{n}\frac{a^2}{(s)^2}\frac{s-1}{n-1}\frac{n-s}{n-2}\sum_{i=1}^n\left\|\hat{x}_i^t\right\|^2 + p\frac{s}{n}\frac{a^2}{(s)^2}\frac{s-1}{n-1}\frac{s-2}{n-2}\sum_{i=1}^n\left\|\sum_{j=1}^n\hat{x}_j^t - \hat{x}_i^t\right\|^2 \\
&\quad - 2p\frac{s}{n}\frac{a}{s}\frac{s-1}{n-1}\left(a-\frac{a}{s}\right)\sum_{i=1}^n\left\langle\hat{x}_i^t,\sum_{j=1}^n\hat{x}_j^t - \hat{x}_i^t\right\rangle \\
&= \frac{(n-1)^2}{psn}\sum_{i=1}^n\left\|\hat{x}_i^t\right\|^2 + \frac{(n-1)^2}{ps(s-1)n}\frac{n-s}{n-2}\sum_{i=1}^n\left\|\hat{x}_i^t\right\|^2 \\
&\quad + \frac{1}{ps}\frac{s-2}{s-1}\frac{n-1}{n-2}\left\|\sum_{i=1}^n\hat{x}_i^t\right\|^2 - 2\frac{1}{psn}\frac{s-2}{s-1}\frac{n-1}{n-2}\left\|\sum_{i=1}^n\hat{x}_i^t\right\|^2 \\
&\quad + \frac{1}{psn}\frac{s-2}{s-1}\frac{n-1}{n-2}\sum_{i=1}^n\left\|\hat{x}_i^t\right\|^2 + 2\frac{n-1}{psn}\sum_{i=1}^n\left\|\hat{x}_i^t\right\|^2 - 2\frac{n-1}{psn}\left\|\sum_{i=1}^n\hat{x}_i^t\right\|^2 \\
&= \frac{(n-1)(n+1)}{psn}\sum_{i=1}^n\left\|\hat{x}_i^t\right\|^2 + \frac{(n-1)^2}{ps(s-1)n}\frac{n-s}{n-2}\sum_{i=1}^n\left\|\hat{x}_i^t\right\|^2 \\
&\quad - \frac{n-1}{psn}\frac{s}{s-1}\left\|\sum_{i=1}^n\hat{x}_i^t\right\|^2 + \frac{1}{psn}\frac{s-2}{s-1}\frac{n-1}{n-2}\sum_{i=1}^n\left\|\hat{x}_i^t\right\|^2 \\
&= \frac{(n^2-1)(s-1)(n-2)+(n-1)^2(n-s)+(s-2)(n-1)}{ps(s-1)n(n-2)}\sum_{i=1}^n\left\|\hat{x}_i^t\right\|^2 \\
&\quad - \frac{n-1}{p(s-1)n}\left\|\sum_{i=1}^n\hat{x}_i^t\right\|^2 \\
&= \frac{n-1}{p(s-1)}\sum_{i=1}^n\left\|\hat{x}_i^t\right\|^2 - \frac{n-1}{p(s-1)n}\left\|\sum_{i=1}^n\hat{x}_i^t\right\|^2 \\
&= \frac{n-1}{p(s-1)}\sum_{i=1}^n\left\|\hat{x}_i^t - \frac{1}{n}\sum_{j=1}^n\hat{x}_j^t\right\|^2 .
\end{aligned}
$$

Therefore, (30) holds with

$$\omega = \frac{n-1}{p(s-1)} - 1 \tag{31}$$

and we have $a = 1 + \omega$.

### A.2 From Algorithm 2 to TAMUNA

TAMUNA is a two-loop version of Algorithm 2, where every sequence of local steps followed by a communication step is grouped into a round. One crucial observation about Algorithm 2 is the following: for a client $i \notin \Omega^t$, which does not participate in communication at iteration $t$ with $\theta^t = 1$, its variable $x_i$ gets overwritten by $\bar{x}^t$ anyway (step 12 of Algorithm 2). Therefore, all local steps it performed since its last participation are useless and can be omitted. But at iteration $t$ with $\theta^t = 0$, it is still undecided whether or not a given client will participate in the next communication step at iteration $t' > t$, since $\Omega^{t'}$ has not yet been generated. That is why TAMUNA is written with two loops, so that it is decided at the beginning of the round which clients will communicate at the end of the round. Accordingly, at the beginning of round $r$, a client downloads the current model estimate (step 6 of TAMUNA) only if it participates ($i \in \Omega^{(r)}$), to initialize its sequence of local steps. Otherwise ($i \notin \Omega^{(r)}$), the client is completely idle: neither computation nor downlink or uplink communication is performed in round $r$. Finally, a round consists of a sequence of successive iterations with $\theta^t = 0$ and a last iteration with $\theta^t = 1$ followed by communication. Thus, the number of iterations, or local steps, in a round can be determined directly at the beginning of the round, according to a geometric law. Given these considerations, Algorithm 2 and TAMUNA are equivalent. In TAMUNA, the round and local step indexing is denoted by parentheses, e.g. $(r, \ell)$, to differentiate it clearly from the iteration indexing.

To obtain Theorem 1 from Theorem 3, we first have to reindex the local steps to make the equivalent iteration index $t$ in Algorithm 2 appear, since the rate is with respect to the number of iterations, not rounds, whose size is random. The almost sure convergence statement follows directly from the one in Theorem 3.

Importantly, we want a result related to the variables which are actually computed in TAMUNA, without including virtual variables by the idle clients, which are computed in Algorithm 2 but not in TAMUNA. That is why we express the convergence result with respect to $\bar{x}^t$, which relates only to the variables of active clients; also, $\bar{x}^t$ is the model estimate known by the server whenever communication occurs, which matters at the end. Note the bar in $\overline{\Psi}$ in (6) to differentiate it from $\Psi$ in (22). Thus, we continue the analysis of Algorithms 2 and 3 in Section A, with same definitions and notations. Let $t \geq 0$. If $\theta^t = 0$, we choose $\Omega^t \subset [n]$ of size $c$ uniformly at random and a random binary mask $\mathbf{q}^t = (q_i^t)_{i \in \Omega^t} \in \mathbb{R}^{d \times c}$, and we define $\bar{x}^t := \frac{1}{s} \sum_{j \in \Omega^t} \mathcal{C}_j^t(\hat{x}_j^t)$ (in Theorem 1, for simplicity, $\Omega^t$ and $\mathbf{q}^t$ are the ones that will be used at the end of the round; this choice is valid as it does not depend on the past). $\Omega^t$, $\mathbf{q}^t$ and $\bar{x}^t$ are already defined if $\theta^t = 1$. We want to study $\mathbb{E}\left[\left\|\bar{x}^t - x^\star\right\|^2 \mid \mathcal{F}^t\right]$, where the expectation is with respect to $\Omega^t$ and $\mathbf{q}^t$, whatever $\theta^t$. Using the derivations already obtained,

$$
\begin{aligned}
n\mathbb{E}\left[\left\|\bar{x}^t - x^\star\right\|^2 \mid \mathcal{F}^t\right] &= \left\|\hat{\mathbf{x}}^t - \mathbf{x}^\star\right\|^2 - \left\|W\hat{\mathbf{x}}^t\right\|^2 + \nu\left\|W\hat{\mathbf{x}}^t\right\|^2 \\
&= \left\|\mathbf{w}^t - \mathbf{w}^\star\right\|^2 - \gamma^2\left\|\mathbf{h}^t - \mathbf{h}^\star\right\|^2 + 2\gamma\langle\hat{\mathbf{x}}^t - \mathbf{x}^\star, \hat{\mathbf{h}}^{t+1} - \mathbf{h}^\star\rangle \\
&\quad + (2p\chi + \nu - 1)\left\|W\hat{\mathbf{x}}^t\right\|^2 \\
&\leq \left\|\mathbf{w}^t - \mathbf{w}^\star\right\|^2 - \gamma^2\left\|\mathbf{h}^t - \mathbf{h}^\star\right\|^2 + 2\gamma\langle\hat{\mathbf{x}}^t - \mathbf{x}^\star, \hat{\mathbf{h}}^{t+1} - \mathbf{h}^\star\rangle \\
&\quad + \left(2p\chi - p(1 - \nu)\right)\left\|W\hat{\mathbf{x}}^t\right\|^2 .
\end{aligned}
$$

Hence,

$$
\begin{aligned}
\frac{n}{\gamma}\mathbb{E}&\left[\left\|\bar{x}^t - x^\star\right\|^2 \mid \mathcal{F}^t\right] + \frac{\gamma(1+\omega)}{p\chi}\mathbb{E}\left[\left\|\mathbf{h}^{t+1} - \mathbf{h}^\star\right\|^2 \mid \mathcal{F}^t\right] \\
&\leq \frac{1}{\gamma}\left\|\mathbf{w}^t - \mathbf{w}^\star\right\|^2 + \frac{\gamma(1+\omega)}{p\chi}\left(1 - \frac{p\chi}{1+\omega}\right)\left\|\mathbf{h}^t - \mathbf{h}^\star\right\|^2
\end{aligned}
$$

and

$$\frac{n}{\gamma}\mathbb{E}\left[\left\|\bar{x}^t - x^\star\right\|^2 \mid \mathcal{F}_0^t\right] + \frac{\gamma(1+\omega)}{p\chi}\mathbb{E}\left[\left\|\mathbf{h}^{t+1} - \mathbf{h}^\star\right\|^2 \mid \mathcal{F}_0^t\right]$$
$$\leq \max\left((1-\gamma\mu)^2, (\gamma L - 1)^2, 1 - p^2\chi\frac{s-1}{n-1}\right)\Psi^t + \gamma\sigma^2.$$

Using the tower rule,

$$\frac{n}{\gamma}\mathbb{E}\left[\left\|\bar{x}^t - x^\star\right\|^2\right] + \frac{\gamma(1+\omega)}{p\chi}\mathbb{E}\left[\left\|\mathbf{h}^{t+1} - \mathbf{h}^\star\right\|^2\right] \leq \tau^t\Psi^0 + \frac{\gamma\sigma^2}{1-\tau}.$$

Since in TAMUNA, $x_1^0 = \cdots = x_n^0 = \bar{x}^0 = \bar{x}^{(0)}$, $\overline{\Psi}^0 = \Psi^0$. This concludes the proof of Theorem 1.

## B   Proof of Theorem 2

We suppose that the assumptions in Theorem 2 hold. $s$ is set as the maximum of three values. Let us consider these three cases.

1) Suppose that $s = 2$. Since $2 = s \geq \lfloor \alpha c \rfloor$ and $2 = s \geq \lfloor \frac{c}{d} \rfloor$, we have $\alpha \leq \frac{3}{c}$ and $1 \leq \frac{3d}{c}$. Hence,

$$\mathcal{O}\left(\sqrt{\frac{n\kappa}{s}} + \frac{n}{s}\right)\left(\frac{sd}{c} + 1 + \alpha d\right)$$
$$= \mathcal{O}\left(\sqrt{n\kappa} + n\right)\left(\frac{d}{c} + \frac{d}{c} + \frac{d}{c}\right)$$
$$= \mathcal{O}\left(d\frac{\sqrt{n\kappa}}{c} + d\frac{n}{c}\right). \tag{32}$$

2) Suppose that $s = \lfloor \frac{c}{d} \rfloor$. Then $\frac{sd}{c} \leq 1$. Since $s \geq \lfloor \alpha c \rfloor$ and $\lfloor \frac{c}{d} \rfloor = s \geq 2$, we have $\alpha c \leq s + 1 \leq \frac{c}{d} + 1$ and $\frac{d}{c} \leq \frac{1}{2}$, so that $\alpha d \leq 1 + \frac{d}{c} \leq 2$. Hence,

$$\mathcal{O}\left(\sqrt{\frac{n\kappa}{s}} + \frac{n}{s}\right)\left(\frac{sd}{c} + 1 + \alpha d\right)$$
$$= \mathcal{O}\left(\sqrt{\frac{n\kappa}{s}} + \frac{n}{s}\right).$$

Since $2s \geq \frac{c}{d}$, we have $\frac{1}{s} \leq \frac{2d}{c}$ and

$$\mathcal{O}\left(\sqrt{\frac{n\kappa}{s}} + \frac{n}{s}\right)\left(\frac{sd}{c} + 1 + \alpha d\right)$$
$$= \mathcal{O}\left(\sqrt{d}\sqrt{\frac{n\kappa}{c}} + d\frac{n}{c}\right). \tag{33}$$

3) Suppose that $s = \lfloor \alpha c \rfloor$. This implies $\alpha > 0$. Then $s \leq \alpha c$. Also, $2s \geq \alpha c$ and $\frac{1}{s} \leq \frac{2}{\alpha c}$. Since $s = \lfloor \alpha c \rfloor \geq \lfloor \frac{c}{d} \rfloor$, we have $\alpha c + 1 \geq \frac{c}{d}$ and $1 \leq \alpha d + \frac{d}{c}$. Since $s = \lfloor \alpha c \rfloor \geq 2$, we have $\frac{1}{c} \leq \frac{\alpha}{2}$ and $1 \leq 2\alpha d$. Hence,

$$\mathcal{O}\left(\sqrt{\frac{n\kappa}{s}} + \frac{n}{s}\right)\left(\frac{sd}{c} + 1 + \alpha d\right)$$
$$= \mathcal{O}\left(\sqrt{\frac{n\kappa}{\alpha c}} + \frac{n}{\alpha c}\right)(\alpha d + \alpha d + \alpha d)$$
$$= \mathcal{O}\left(\sqrt{\alpha}d\sqrt{\frac{n\kappa}{c}} + d\frac{n}{c}\right). \tag{34}$$

By adding up the three upper bounds (32), (33), (34), we obtain the upper bound in (15).

## C   Sublinear convergence in the convex case

In this section only, we remove the hypothesis of strong convexity: the functions $f_i$ are only assumed to be convex and $L$-smooth, and we suppose that a solution $x^\star \in \mathbb{R}^d$ to (1) exists. Also, for simplicity, we only consider the case of exact gradients ($\sigma = 0$). Then we have sublinear ergodic convergence:

**Theorem 4** (sublinear convergence). *In Algorithm 2 suppose that $\sigma = 0$ and that*

$$0 < \gamma < \frac{2}{L} \quad and \quad 0 < \chi < \frac{n(s-1)}{s(n-1)} \in \left(\frac{1}{2}, 1\right]. \tag{35}$$

*For every $i = 1, \ldots, n$ and $T \geq 0$, let*

$$\tilde{x}_i^T := \frac{1}{T+1} \sum_{t=0}^{T} x_i^t. \tag{36}$$

*Then*

$$\mathbb{E}\left[\left\|\nabla f(\tilde{x}_i^T)\right\|^2\right] = \mathcal{O}\left(\frac{1}{T}\right). \tag{37}$$

*Proof.* A solution $x^\star \in \mathbb{R}^d$ to (1), which is supposed to exist, satisfies $\nabla f(x^\star) = \frac{1}{n} \sum_{i=1}^{n} \nabla f_i(x^\star) = 0$. $x^\star$ is not necessarily unique but $h_i^\star := \nabla f_i(x^\star)$ is unique.

We define the Bregman divergence of a $L$-smooth convex function $g$ at points $x, x' \in \mathbb{R}^d$ as $D_g(x, x') := g(x) - g(x') - \langle \nabla g(x'), x - x' \rangle \geq 0$. We have $D_g(x, x') \geq \frac{1}{2L} \|\nabla g(x) - \nabla g(x')\|^2$. We note that for every $x \in \mathbb{R}^d$ and $i = 1, \ldots, n$, $D_{f_i}(x, x^\star)$ is the same whatever the solution $x^\star$.

For every $t \geq 0$, we define the Lyapunov function

$$\Psi^t := \frac{1}{\gamma} \sum_{i=1}^{n} \left\|x_i^t - x^\star\right\|^2 + \frac{\gamma}{p^2\chi} \frac{n-1}{s-1} \sum_{i=1}^{n} \left\|h_i^t - h_i^\star\right\|^2, \tag{38}$$

Starting from (26), we have, for every $t \geq 0$,

$$\mathbb{E}\left[\Psi^{t+1} \mid \mathcal{F}^t\right] = \frac{1}{\gamma} \sum_{i=1}^{n} \mathbb{E}\left[\left\|x_i^{t+1} - x^\star\right\|^2 \mid \mathcal{F}^t\right] + \frac{\gamma}{p^2\chi} \frac{n-1}{s-1} \sum_{i=1}^{n} \mathbb{E}\left[\left\|h_i^{t+1} - h_i^\star\right\|^2 \mid \mathcal{F}^t\right]$$

$$\leq \frac{1}{\gamma} \sum_{i=1}^{n} \left\|\left(x_i^t - \gamma \nabla f_i(x_i^t)\right) - \left(x^\star - \gamma \nabla f_i(x^\star)\right)\right\|^2$$

$$+ \left(\frac{\gamma}{p^2\chi} \frac{n-1}{s-1} - \gamma\right) \sum_{i=1}^{n} \left\|h_i^t - h_i^\star\right\|^2 + \frac{p}{\gamma}(\chi - 1 + \nu) \sum_{i=1}^{n} \left\|\hat{x}_i^t - \frac{1}{n} \sum_{j=1}^{n} \hat{x}_j^t\right\|^2,$$

with

$$\left\|\left(x_i^t - \gamma \nabla f_i(x_i^t)\right) - \left(x^\star - \gamma \nabla f_i(x^\star)\right)\right\|^2 = \left\|x_i^t - x^\star\right\|^2 - 2\gamma\langle \nabla f_i(x_i^t) - \nabla f_i(x^\star), x_i^t - x^\star \rangle$$

$$+ \gamma^2 \left\|\nabla f_i(x_i^t) - \nabla f_i(x^\star)\right\|^2$$

$$\leq \left\|x_i^t - x^\star\right\|^2 - (2\gamma - \gamma^2 L)\langle \nabla f_i(x_i^t) - \nabla f_i(x^\star), x_i^t - x^\star \rangle,$$

where the second inequality follows from cocoercivity of the gradient. Moreover, for every $x, x'$, $D_{f_i}(x, x') \leq \langle \nabla f_i(x) - \nabla f_i(x'), x - x' \rangle$. Therefore,

$$\mathbb{E}\left[\Psi^{t+1} \mid \mathcal{F}^t\right] \leq \Psi^t - (2 - \gamma L) \sum_{i=1}^{n} D_{f_i}(x_i^t, x^\star)$$

$$- \gamma \sum_{i=1}^{n} \left\|h_i^t - h_i^\star\right\|^2 + \frac{p}{\gamma}(\chi - 1 + \nu) \sum_{i=1}^{n} \left\|\hat{x}_i^t - \frac{1}{n} \sum_{j=1}^{n} \hat{x}_j^t\right\|^2.$$

Telescoping the sum and using the tower rule of expectations, we get summability over $t$ of the three negative terms above: for every $T \geq 0$, we have

$$(2 - \gamma L) \sum_{i=1}^{n} \sum_{t=0}^{T} \mathbb{E}\big[D_{f_i}(x_i^t, x^\star)\big] \leq \Psi^0 - \mathbb{E}\big[\Psi^{T+1}\big] \leq \Psi^0, \tag{39}$$

$$\gamma \sum_{i=1}^{n} \sum_{t=0}^{T} \mathbb{E}\Big[\big\|h_i^t - h_i^\star\big\|^2\Big] \leq \Psi^0 - \mathbb{E}\big[\Psi^{T+1}\big] \leq \Psi^0, \tag{40}$$

$$\frac{p}{\gamma}(1 - \nu - \chi) \sum_{i=1}^{n} \sum_{t=0}^{T} \mathbb{E}\left[\left\|\hat{x}_i^t - \frac{1}{n}\sum_{j=1}^{n}\hat{x}_j^t\right\|^2\right] \leq \Psi^0 - \mathbb{E}\big[\Psi^{T+1}\big] \leq \Psi^0. \tag{41}$$

Taking ergodic averages and using convexity of the squared norm and of the Bregman divergence, we can now get $\mathcal{O}(1/T)$ rates. We use a tilde to denote averages over the iterations so far. That is, for every $i = 1, \ldots, n$ and $T \geq 0$, we define

$$\tilde{x}_i^T := \frac{1}{T+1} \sum_{t=0}^{T} x_i^t$$

and

$$\tilde{x}^T := \frac{1}{n} \sum_{i=1}^{n} \tilde{x}_i^T.$$

The Bregman divergence is convex in its first argument, so that, for every $T \geq 0$,

$$\sum_{i=1}^{n} D_{f_i}(\tilde{x}_i^T, x^\star) \leq \sum_{i=1}^{n} \frac{1}{T+1} \sum_{t=0}^{T} D_{f_i}(x_i^t, x^\star).$$

Combining this inequality with (39) yields, for every $T \geq 0$,

$$(2 - \gamma L) \sum_{i=1}^{n} \mathbb{E}\big[D_{f_i}(\tilde{x}_i^T, x^\star)\big] \leq \frac{\Psi^0}{T+1}. \tag{42}$$

Similarly, for every $i = 1, \ldots, n$ and $T \geq 0$, we define

$$\tilde{h}_i^T := \frac{1}{T+1} \sum_{t=0}^{T} h_i^t$$

and we have, for every $T \geq 0$,

$$\sum_{i=1}^{n} \big\|\tilde{h}_i^T - h_i^\star\big\|^2 \leq \sum_{i=1}^{n} \frac{1}{T+1} \sum_{t=0}^{T} \big\|h_i^t - h_i^\star\big\|^2.$$

Combining this inequality with (40) yields, for every $T \geq 0$,

$$\gamma \sum_{i=1}^{n} \mathbb{E}\Big[\big\|\tilde{h}_i^T - h_i^\star\big\|^2\Big] \leq \frac{\Psi^0}{T+1}. \tag{43}$$

Finally, for every $i = 1, \ldots, n$ and $T \geq 0$, we define

$$\tilde{\hat{x}}_i^T := \frac{1}{T+1} \sum_{t=0}^{T} \hat{x}_i^t$$

and

$$\tilde{\tilde{x}}^T := \frac{1}{n} \sum_{i=1}^{n} \tilde{\tilde{x}}_i^T,$$

and we have, for every $T \geq 0$,

$$\sum_{i=1}^{n} \left\| \tilde{\tilde{x}}_i^T - \tilde{\tilde{x}}^T \right\|^2 \leq \sum_{i=1}^{n} \frac{1}{T+1} \sum_{t=0}^{T} \left\| \hat{x}_i^t - \frac{1}{n} \sum_{j=1}^{n} \hat{x}_j^t \right\|^2.$$

Combining this inequality with (41) yields, for every $T \geq 0$,

$$\frac{p}{\gamma}(1 - \nu - \chi) \sum_{i=1}^{n} \mathbb{E}\left[ \left\| \tilde{\tilde{x}}_i^T - \tilde{\tilde{x}}^T \right\|^2 \right] \leq \frac{\Psi^0}{T+1}. \tag{44}$$

Next, we have, for every $i = 1, \ldots, n$ and $T \geq 0$,

$$\left\| \nabla f(\tilde{x}_i^T) \right\|^2 \leq 2 \left\| \nabla f(\tilde{x}_i^T) - \nabla f(\tilde{x}^T) \right\|^2 + 2 \left\| \nabla f(\tilde{x}^T) \right\|^2$$
$$\leq 2L^2 \left\| \tilde{x}_i^T - \tilde{x}^T \right\|^2 + 2 \left\| \nabla f(\tilde{x}^T) \right\|^2. \tag{45}$$

Moreover, for every $T \geq 0$ and solution $x^\star$ to (1),

$$\left\| \nabla f(\tilde{x}^T) \right\|^2 = \left\| \nabla f(\tilde{x}^T) - \nabla f(x^\star) \right\|^2$$
$$\leq \frac{1}{n} \sum_{i=1}^{n} \left\| \nabla f_i(\tilde{x}^T) - \nabla f_i(x^\star) \right\|^2$$
$$\leq \frac{2}{n} \sum_{i=1}^{n} \left\| \nabla f_i(\tilde{x}^T) - \nabla f_i(\tilde{x}_i^T) \right\|^2 + \frac{2}{n} \sum_{i=1}^{n} \left\| \nabla f_i(\tilde{x}_i^T) - \nabla f_i(x^\star) \right\|^2$$
$$\leq \frac{2L^2}{n} \sum_{i=1}^{n} \left\| \tilde{x}_i^T - \tilde{x}^T \right\|^2 + \frac{4L}{n} \sum_{i=1}^{n} D_{f_i}(\tilde{x}_i^T, x^\star). \tag{46}$$

There remains to control the terms $\left\| \tilde{x}_i^T - \tilde{x}^T \right\|^2$: we have, for every $T \geq 0$,

$$\sum_{i=1}^{n} \left\| \tilde{x}_i^T - \tilde{x}^T \right\|^2 \leq 2 \sum_{i=1}^{n} \left\| (\tilde{x}_i^T - \tilde{x}^T) - (\tilde{\tilde{x}}_i^T - \tilde{\tilde{x}}^T) \right\|^2 + 2 \sum_{i=1}^{n} \left\| \tilde{\tilde{x}}_i^T - \tilde{\tilde{x}}^T \right\|^2$$
$$\leq 2 \sum_{i=1}^{n} \left\| \tilde{x}_i^T - \tilde{\tilde{x}}_i^T \right\|^2 + 2 \sum_{i=1}^{n} \left\| \tilde{\tilde{x}}_i^T - \tilde{\tilde{x}}^T \right\|^2. \tag{47}$$

For every $i = 1, \ldots, n$ and $t \geq 0$,

$$\hat{x}_i^t = x_i^t - \gamma \left( \nabla f_i(x_i^t) - h_i^t \right)$$

so that, for every $i = 1, \ldots, n$ and $T \geq 0$,

$$\tilde{x}_i^T - \tilde{\tilde{x}}_i^T = \gamma \frac{1}{T+1} \sum_{t=0}^{T} \nabla f_i(x_i^t) - \gamma \tilde{h}_i^T$$

and

$$\left\| \tilde{x}_i^T - \tilde{\tilde{x}}_i^T \right\|^2 = \gamma^2 \left\| \frac{1}{T+1} \sum_{t=0}^{T} \nabla f_i(x_i^t) - \tilde{h}_i^T \right\|^2$$
$$\leq 2\gamma^2 \frac{1}{T+1} \sum_{t=0}^{T} \left\| \nabla f_i(x_i^t) - \nabla f_i(x^\star) \right\|^2 + 2\gamma^2 \left\| \tilde{h}_i^T - h_i^\star \right\|^2$$
$$\leq 4L\gamma^2 \frac{1}{T+1} \sum_{t=0}^{T} D_{f_i}(x_i^t, x^\star) + 2\gamma^2 \left\| \tilde{h}_i^T - h_i^\star \right\|^2. \tag{48}$$

Combining (45), (46), (47), (48), we get, for every $T \geq 0$,

$$
\sum_{i=1}^{n} \left\| \nabla f(\tilde{x}_i^T) \right\|^2 \leq 2L^2 \sum_{i=1}^{n} \left\| \tilde{x}_i^T - \tilde{x}^T \right\|^2 + 2n \left\| \nabla f(\tilde{x}^T) \right\|^2
$$

$$
\leq 2L^2 \sum_{i=1}^{n} \left\| \tilde{x}_i^T - \tilde{x}^T \right\|^2 + 2L^2 \sum_{i=1}^{n} \left\| \tilde{x}_i^T - \tilde{x}^T \right\|^2 + 4L \sum_{i=1}^{n} D_{f_i}(\tilde{x}_i^T, x^\star)
$$

$$
= 4L^2 \sum_{i=1}^{n} \left\| \tilde{x}_i^T - \tilde{x}^T \right\|^2 + 4L \sum_{i=1}^{n} D_{f_i}(\tilde{x}_i^T, x^\star)
$$

$$
\leq 8L^2 \sum_{i=1}^{n} \left\| \tilde{x}_i^T - \tilde{\tilde{x}}_i^T \right\|^2 + 8L^2 \sum_{i=1}^{n} \left\| \tilde{\tilde{x}}_i^T - \tilde{x}^T \right\|^2 + 4L \sum_{i=1}^{n} D_{f_i}(\tilde{x}_i^T, x^\star)
$$

$$
\leq 32L^3 \gamma^2 \frac{1}{T+1} \sum_{i=1}^{n} \sum_{t=0}^{T} D_{f_i}(x_i^t, x^\star) + 16L^2 \gamma^2 \sum_{i=1}^{n} \left\| \tilde{h}_i^T - h_i^\star \right\|^2
$$

$$
+ 8L^2 \sum_{i=1}^{n} \left\| \tilde{\tilde{x}}_i^T - \tilde{x}^T \right\|^2 + 4L \sum_{i=1}^{n} D_{f_i}(\tilde{x}_i^T, x^\star).
$$

Taking the expectation and using (39), (43), (44) and (42), we get, for every $T \geq 0$,

$$
\sum_{i=1}^{n} \mathbb{E}\left[ \left\| \nabla f(\tilde{x}_i^T) \right\|^2 \right] \leq 32L^3 \gamma^2 \frac{1}{T+1} \sum_{i=1}^{n} \sum_{t=0}^{T} \mathbb{E}\left[ D_{f_i}(x_i^t, x^\star) \right]
$$

$$
+ 16L^2 \gamma^2 \sum_{i=1}^{n} \mathbb{E}\left[ \left\| \tilde{h}_i^T - h_i^\star \right\|^2 \right]
$$

$$
+ 8L^2 \sum_{i=1}^{n} \mathbb{E}\left[ \left\| \tilde{\tilde{x}}_i^T - \tilde{x}^T \right\|^2 \right] + 4L \sum_{i=1}^{n} \mathbb{E}\left[ D_{f_i}(\tilde{x}_i^T, x^\star) \right].
$$

$$
\leq \frac{32L^3 \gamma^2}{2 - \gamma L} \frac{\Psi_0}{T+1} + 16L^2 \gamma \frac{\Psi_0}{T+1} + \frac{8L^2 \gamma}{p(1 - \nu - \chi)} \frac{\Psi_0}{T+1} + \frac{4L}{2 - \gamma L} \frac{\Psi_0}{T+1}
$$

$$
= \left[ \frac{32L^3 \gamma^2 + 4L}{2 - \gamma L} + 16L^2 \gamma + \frac{8L^2 \gamma}{p(1 - \nu - \chi)} \right] \frac{\Psi_0}{T+1}.
$$

$\square$

Hence, with $\gamma = \Theta\left( \frac{p}{L} \sqrt{\frac{c}{n}} \right)$, $\chi$ satisfying $\delta \leq \chi \leq 1 - \nu - \delta$ for some $\delta > 0$, and $h_i^0 = \nabla f_i(x^0)$, for every $i \in [n]$, then for every $\epsilon > 0$, we have

$$
\sum_{i=1}^{n} \mathbb{E}\left[ \left\| \nabla f(\tilde{x}_i^T) \right\|^2 \right] \leq \epsilon \tag{49}
$$

after

$$
\mathcal{O}\left( \frac{L^2}{p} \sqrt{\frac{n}{c}} \frac{\left\| \mathbf{x}^0 - \mathbf{x}^\star \right\|^2}{\epsilon} \right) \tag{50}
$$

iterations and

$$
\mathcal{O}\left( L^2 \sqrt{\frac{n}{c}} \frac{\left\| \mathbf{x}^0 - \mathbf{x}^\star \right\|^2}{\epsilon} \right) \tag{51}
$$

communication rounds.

We note that LT does not yield any acceleration: the communication complexity is the same whatever $p$. CC is effective, however, since we communicate much less than $d$ floats during every communication round.

This convergence result applies to Algorithm 2. $\tilde{x}_i^T$ in (36) is an average of all $x_i^t$, including the ones for clients not participating in the next communication round. The result still applies to TAMUNA, with, for

every $i \in [n]$, $\tilde{x}_i^T$ defined as the average of the $x_i^{(r,\ell)}$ which are actually computed, since this is a random subsequence of all $x_i^t$.

