# OpenReview forum: "TAMUNA: Doubly Accelerated Federated Learning with Local Training, Compression, and Partial Participation"
_TMLR — Rejected by TMLR_

### Review · Reviewer_L9uH · 2023-08-29

**Summary Of Contributions:**

This work considers communication compression, partial participation in federated learning, and upload/download difference to save the communication cost.

**Audience:**

Yes

**Broader Impact Concerns:**

This is mainly a theoretical work and I don't have broader impact concerns.

**Claims And Evidence:**

No

**Requested Changes:**

See above.

**Strengths And Weaknesses:**

Strengths:
1. Saving communication cost in federated learning and this work considers upload/download communication difference, which is a practical view.
2. It conducts extensive theoretical analysis for the convergence rate and communication cost.
3. Experiments on logistic regression shows the improvement of TAMUNA over 5GCS and Scaffnew, which have the same communication complexity.

Weaknesses:
1. Suitability: since this work mainly considers the communication cost and the difference between upload/download, I think it might be more suitable for communication conferences, such as GLOBECOM or INFOCOM.
2. Sec 1.3.2 is missing an important citation by Li et al. (https://arxiv.org/pdf/1907.02189.pdf)
3. Since DIANA-PP allows for partial participation and compression, what is the difference between TAMUNA and DIANA-PP?
4. Table 1 $\mathcal{O}$ should have a tilde on top. Also could you explain $d, \kappa, c, n$ in the caption as well?
5. Table 2: Scaffnew, 5GCS, and TAMUNA have the same total complexity. Is there a theoretical lower bound for this? Are they optimal? Also the assumption $\alpha = 0$ is not practical. Could you give some situations for when this happens? Or is there any related work?
6. Usage of "don't$ is a bit casual
7. Page 6: "We also stress that our goal is to deepen... " I don't see why proposing another LT + PP + CC algorithm would deepen our understanding.
8. The notations are quite confusing. $\mathcal{L}$ is usually used for loss functions but it is used as the number of local steps here.
9. Before Sec 1.4.2: why does the sum of the control variates remain zero? Why is this important?
10. Sec 1.4.2: Condat et al. (2022) typo
11. Page 8: why is there no way to enable PP in Scaffnew? I think DIANA-PP did this?
12. The equation before Sec 2 is inconsistent with the main result. It can only be an approximate result.
13. Theorem 1: could you explain "a geometric law of mean $p^{-1}$? Could you explain how you found the Lyapunov function? Also the final result converges to a neighborhood depending on the variance, and this point should be made clear.
14. The notations $\ell$, $\bar{x}^{(t)}$ and $\bar{x}^{(r, \mathcal{L}^{(r)})}$ are really confusing.
15. Before Sec 3, the difference between TAMUNA and Scaffold should be highlighted.
16. Is the sparsify index explained somewhere in the paper? Could you explain more?
17. The experiments only considers the cases when $\alpha$ is zero or very small, and studies only logistic regression. More practical settings should be considered, like non-convex deep learning experiments or more practical $\alpha$.
18. Table 3: there are 3 stepsizes, $\gamma, \eta$ and $\chi$. What are the differences? Also not all notations are included in the table.
19. Theorem 1 is missing assumptions, like convexity.

---

> ### Author Response · Authors · 2023-11-02
> **Response 1/2**
>
> Thank you for your thorough and positive evaluation of our work.
>
> > 1. Suitability: since this work mainly considers the communication cost and the differences between upload/download, I think it might be more suitable for communication conferences, such as GLOBECOM or INFOCOM.
>
> Communication is the bottleneck in federated learning (FL), decentralized optimization, and many other areas at the core of machine learning (ML). There are workshops dedicated to FL at the leading ML conferences such as ICML and NeurIPS.
>
> > 2. Sec 1.3.2 is missing an important citation by Li et al.
>
> Yes, we forgot to include this citation, thank you for reminding us. It has been added in the revised paper.
>
> > 3. Since DIANA-PP allows for partial participation and compression, what is the difference between TAMUNA and DIANA-PP?
>
> As indicated in Table 1, DIANA-PP does not feature local training. That is, vectors are communicated at every iteration, or equivalently after every gradient computation step. Thus, with full participation and $n\geq d$, its uplink communication complexity is $\tilde{O}(\kappa + d)$, thanks to compression, which is much better than $\tilde{O}(\kappa d)$ with standard gradient descent. But full dimensional vectors are broadcast at every iteration of DIANA-PP, so its downlink complexity remains $\tilde{O}(\kappa d)$. By contrast, in all regimes, TAMUNA benefits from the acceleration due to local training and has a complexity depending on $\sqrt{\kappa}$ instead of $\kappa$. TAMUNA also enjoys the second acceleration in $\sqrt{d}$ in some regimes.
>
> > 4. Table 1 should have a tilde on top.
>
> There is a tile on all big-Os. Do you see a place where the tilde is missing? In Table 2 footnote (a) there is no tilde, since we compare the leading factors. We added big-Os in Table 1 (a) to make these two footnotes consistent.
>
> > Also could you explain in the caption as well?
>
> These notations are explained in Table 3. We have added a mention to Table 3 in the captions of Tables 1 and 2.
>
> > 5. Table 2: Scaffnew, 5GCS and TAMUNA have the same total complexity
>
> This is not the case. This is true only if $\alpha$ is large, but as we show in the experiments, there is still some practical improvement. In the worst case where $\alpha=1$, compression is disabled, and in case of full participation TAMUNA = Scaffnew with complexity $\tilde{O}(d \sqrt{\kappa})$. TAMUNA is still new in case of partial participation. In other words, the second acceleration mechanism of decreasing the dependency on $d$ through compression is effective for every $\alpha$, and is disabled when $\alpha=1$. $\alpha=0$ is the best case where we get the $\sqrt{d}$ complexity instead of $d$, but again, practical acceleration applies for every $\alpha \in [0,1)$.
>
> > Is there a theoretical lower bound for this?
>
> We are not aware of any existing lower bound.
>
> > 6. Usage of "don't" is a bit casual
>
> We have replaced the "don't" by "do not"
>
> > 7. I don't see why proposing another LT + PP + CC algorithm would deepen our understanding.
>
> TAMUNA is the first algorithm with LT + PP + CC converging linearly to the exact solution. We are not aware of any other algorithm with these 3 features. By showing that this can be achieved, and how, we deepen our understanding of communication-efficient distributed optimization.
>
> >8. $\mathcal{L}$ is usually used for loss functions.
>
> Our results are not stated in terms of objective values so the confusion should not occur. We think that L for "Local" makes sense but L is already used for the Lipschitz constant, as is common. The letters k,h,p,q,r,s,t,x,c,n,i,d... are already used, which leaves no good choice and a greek letter would look weird.
>
> > 9. why does the sum of the control variates remain zero?  Why is this important?
>
> The active control variates $h_i$ are updated using vectors whose sum is zero, because these are vectors whose average is subtracted to all of them. This property means that the control variates remain in the orthogonal complement of the consensus line { $x_1=\cdots=x_n$ }. It is crucial that the sum of the random errors remains zero, because this way the errors do not cause any drift on the model. They slow down convergence but do not jeopardize it. Without this property, errors might propagate throughout the iterations and make the algorithm diverge.
>
> > 10. Condat et al. (2022) typo
>
> What is the typo? In any case we have rephrased the sentence to avoid the double use of "Condat et al." for the reference and the authors.
>
> > 11. Page 8: why is there no way to enable PP in Scaffnew? I think DIANA-PP did this?
>
> DIANA proposed in 2019 and Scaffnew proposed in 2022 are very different algorithms. As replied above to your point 3., DIANA has compression and no local training. Scaffnew has local training and no compression. Compression and local training are clearly distinct  mechanisms. TAMUNA is a way to enable PP in Scaffnew, there might be other ways but TAMUNA is the first one to be introduced.

---

> ### Author Response · Authors · 2023-11-02
> **Response 2/2**
>
> > 12. The equation before Sec 2 is inconsistent with the main result. It can only be an approximate result.
>
> We think the result is correct. If you replace $c$ by $n$ and $n\geq d$ in eq. 16 of Corollary 1, you obtain the stated result.
>
> > 13. Theorem 1: could you explain "a geometric law of mean $p^{-1}$"?
>
> The explicit formula of the probability is given right after this phrase.
>
> > Could you explain how you found the Lyapunov function?
>
> The conceptual mechanisms in TAMUNA are novel and inspired by the recent paper of Condat and Richtárik, "RandProx: Primal-dual optimization algorithms with randomized proximal updates", ICLR 2023, where a general variance-reduced strategy is presented to randomize the dual update of primal-dual algorithms. In our setting, the dual update corresponds to the intermittent update of the control variates of the participating clients. This observation has been added to the text. We found the Lyapunov function by analogy with the one in this paper. In short, the factor in front of the second squared norm in (6) is chosen so that the inner products, which appear when expanding the involved squared norms, cancel out.
>
> > Also the final result  converges to a neighborhood depending on the variance, and this point should be made clear.
>
> Yes, at this point there is no internal variance reduction to cancel the noise due to the stochastic gradients. This is left for future work, as we wrote in the conclusion. We have modified the title of Theorem 1.
>
> > 14. The notations $\ell$, $x^t$ and $x^{(r,\ell)}$ are really confusing.
>
> There are 2 notations : $t$ is the iteration index for the loopless equivalent algorithm; the pair $(r,\ell)$ indexes the epochs and the local steps inside them for the 2-level algorithm TAMUNA.
>
> > 15. the difference between TAMUNA and Scaffold should be highlighted.
>
> They are quite different algorithms. In particular, there is no compression in Scaffold. Without compression and in case of full participation, TAMUNA reverts to Scaffnew. The name Scaffnew is a play on words to emphasize that it is an improvement over Scaffold, because it has accelerated convergence, unlike Scaffold.
>
> > 16. Is the sparsify index explained somewhere in the paper?
>
> It is described in Figure 1.
>
> > 17. More practical settings should be considered, like non-convex deep learning experiments
>
> We consider the setting of convex optimization. Many practical problems, in particular in deep learning, are not convex, and we currently don't know how to analyze the convergence of TAMUNA in the nonconvex setting. TAMUNA extends Scaffnew presented at ICML 2022, which to the best of our knowledge has only been studied in the convex setting. There exist non-accelerated algorithms and our contribution is to demonstrate acceleration for the first time, albeit in the convex setting. We have added a sentence in the conclusion to reflect this.
>
> > 18. There are 3 stepsizes, $\gamma$, $\eta$, $\chi$. What are the differences?
>
> $\gamma$ is the stepsize used in the gradient descent steps, a.k.a. learning rate.
> $\eta$ is the dual stepsize used to scale the update of the control variates, in step 14 of TAMUNA
> In Theorem 1, $\chi$ is $\eta/p$, as mentioned before (5), which only depends on $n$ and $s$.
> There is no $p$ in TAMUNA, $p$ is defined in Theorem 1 which states convergence if the number of local steps is chosen in a particular probabilistic way that depends on $p$ and $\eta$ is set accordingly.
>
> > 19. Theorem 1 is missing assumptions, like convexity.
>
> The whole paper is in the convex setting, and the assumptions are described at the beginning of the paper in Section 1.1 - Formalism, when defining the problem in eq. (1) and the text right after. The assumptions on the parameters of the algorithm, on the other hand, are stated in the theorems.

---

### Review · Reviewer_Chad · 2023-09-13

**Summary Of Contributions:**

This paper proposes a technique for federated learning, called TAMUNA, which is claimed to be robust to partial participation (PP) by using both local training (LT) and communication compression (CC).

The authors provided some convergence analysis on TAMUNA, and small experimental results.

**Audience:**

Yes

**Broader Impact Concerns:**

.

**Claims And Evidence:**

No

**Requested Changes:**

Suggestions:
- Make a table comparing the different features (convergence rate, communication overhead) of TAMUNA and other baselines?
- Experiments are done for simplest setting (logistic regression problem) only. For improving the practical impact of this paper, I think it is better to include experimental results for more complex tasks.
- In Fig.3, is the shaded region error bar? If so, we usually plot error bar in the range of [\mu-\sigma, \mu+\sigma)], which is not true for Fig.3
- It is better to include conclusion section.


Question:
- Difference between TAMUNA & other related schemes? From Algorithm 1, it is hard to extract the difference at once. I think it is better to mention the difference more precisely in the manuscript.
- Why double acceleration is achievable only for TAMUNA?

**Strengths And Weaknesses:**

Strength
- For simplest setting, TAMUNA has improved performance compared with other baselines.

Weakness
- The conceptual difference between TAMUNA and other baselines is not explicitly mentioned
- Readers can get less insight on why TAMUNA works well and why "double acceleration" occurs.
- Experiments are only provided for simplest setting

---

> ### Author Response · Authors · 2023-11-02
> **Response**
>
> Thank you for your thorough and positive evaluation of our work.
>
> > The conceptual difference between TAMUNA and other baselines is not explicitly mentioned.
>
> We believe the difference between TAMUNA and existing algorithms is explicitly visible in Tables 1 and 2. Conceptually, the mechanisms in TAMUNA are novel and inspired by the recent paper of Condat and Richtárik, "RandProx: Primal-dual optimization algorithms with randomized proximal updates", ICLR 2023, where a general variance-reduced strategy is presented to randomize the dual update of primal-dual algorithms. In our setting, the dual update corresponds to the intermittent update of the control variates of the participating clients. This observation has been added in the text.
>
> > Readers can get less insight on why TAMUNA works well and why "double acceleration" occurs.
>
> Do you mean that it is not clear why TAMUNA works well and why double acceleration occurs? Do you have any suggestions for points that should be made clearer? At the beginning of Section 1.4 we give the formula TAMUNA = (SGD + LT)_Scaffnew + PP + CC. The first acceleration is due to LT like in Scaffnew, the second acceleration is due to compression and is novel.
>
> > Make a table comparing the different features (convergence rate, communication overhead) of TAMUNA and other baselines?
>
> We believe Tables 1 and 2 provide this comparison.
>
> > Experiments are done for simplest setting (logistic regression problem) only. For improving the practical impact of this paper, I think it is better to include experimental results for more complex tasks.
>
> We consider the setting of convex optimization. Many practical problems, in particular in deep learning, are not convex, and we currently don't know how to analyze the convergence of TAMUNA in the nonconvex setting. TAMUNA extends the recent algorithm Scaffnew presented at ICML 2022, which to the best of our knowledge has only been studied in the convex setting. There exist non-accelerated algorithms and our contribution is to demonstrate acceleration for the first time, albeit in the convex setting. We have added a sentence in the conclusion to reflect this.
>
> > In Fig. 3, is the shaded region error bar? If so, we usually plot error bar in the range of [\mu-\sigma, \mu+\sigma)], which is not true for Fig. 3
>
> As written in Section 4, "the shaded area in the plots shows the difference between the maximum and minimum convergence error achieved over these runs. Additionally, the progress of the first run for each algorithm is depicted with a thicker line and markers." Indeed the mean minus the standard deviation can be negative at some points, which gives meaningless plots in log scale.
>
> > It is better to include conclusion section.
>
> We have added a conclusion.

---

### Review · Reviewer_7BWA · 2023-10-22

**Summary Of Contributions:**

The paper presents and analyzes TAMUNA, a new algorithm for federated learning, in which clients perform local optimisation of shared weights on a finite sum minimization problem. In particular, the paper defines and studies the Upload,Download and Total Communication Complexity (UpCom,DownCom, TotalCom) under and assumption that $TotalCom=UpCom+\alpha DownCom$ where the paper focuses on the regime $\alpha \in [0,1]$, mirroring cheaper downloads than uploads (as common in consumer infrastructure). The complexity is defined as the total number of reals sent up/down the wire, and thus lowered by compression (compressed communication,CC), performing more local work between communication rounds (lcoal training, LT) and skipping participation of a client in a round (partial participation, PP).

The core focus of the paper is achieving a guaranteed "double acceleration", in the sense of reducing $TotalCom$ via combining LT,CC and PP, thus reducing the dependence on the condition number $\kappa=\frac{L}{\mu}$ and dimensionality $d$ by a square root factor  (via PP for $\kappa$, via CC for $d$).

The proof is established w.r.t. convergence to a neighbourhood dependent on the gradient noise $\sigma$, improved reduction of the neighbourhood size is not studied as the work focuses on the communication complexity side and speeding up the convergence in the noisy case is left as future work.

The paper studies $\mu$-strongly, $L$- convex finite sum problems (mere convexity is studied in the appendix) and notably does not make any assumptions on data similarity, only a bounded variance on the gradient estimates.

The algorithm proceeeds by selecting a subset of participating nodes, sending the server estimate as an initializer to each worker (at which point a control variate update is performed on the worker, see later),  performing local gradient descent tempered by the current control variate $x_i^{(r,l+1)}=x_i^{(r,l)}-\gamma g_i^{(r,l)}+\gamma h_i^{(r)}$ with $g_i^l$ being the gradient at timestep $l$ and $h_i^r$ the control variate at that round, the worker and the using a pre-arranged randomized compression mask which selects which dimensions of $x$ will be sent to the server for aggregation and finally, the worker updating its control variate with the next value it receives. Notably, this last step can be folded into the reception of the initializer (i.e. for $c$ participating nodes, $x^{r+1}$ is sent to at most $2c$ nodes, the newly participating and the previously participating to update the control variate).

If i understand correctly, the main "trick" is to exploit the separability of expectation and squared euclidean norm and using an unbiased compression algorithm (top-k). Combined this  allows to construct a  form of random coordinate descent w.r.t the local objective being executed (by only updating the control variates where the algorithm made progress towards the local minimum), while maintaining global convergence.

**Audience:**

Yes

**Broader Impact Concerns:**

I think the paper addresses everything.

**Claims And Evidence:**

Yes

**Requested Changes:**

- adding discussion to the reference I found (if relevant, maybe I completely missed the point) + generally further discusison of relevant work seems critical (I am happy to receive an explanation why a bunch of papers are not relevant and why my heuristic is misfiring)
- the wallclock experiment is a nice to have, but I think it would greatly help put the method into a practical context (i.e. "if you have an environment with these latency characteristics, this method will help you X %)
- the question I would simply be curious about but I think adding it to the discussion would be a mini-strengthening of the work
- finally, you have what I assume is a typo on PD page 28 "Telescopic" should probably be "Telescoping" ?

**Strengths And Weaknesses:**

Strengths:

The paper is very clearly written, in particular the math is very legibly presented. While I wasn't able to redo the proofs and check them in detail myself due to time constraints, I expect reviewers more familiar with the material to have an easy time following and checking things, and simply reading them it appeared correct to me.

The paper is also well scoped, doesn't overclaim, and makes a very complete story overall

Weaknesses:

- while there is an extensive treatment of prior art and this is purely a heuristic criticism, about 2/3 of the cited papers share a single prolific author. While this is totally defensible due to an author/group simply finding their and stride dominating a field together, I was able to find https://proceedings.mlr.press/v180/jhunjhunwala22a.html in a brief google scholar trawl which seems relevant, so I think there might be other works that are relevant to discuss outside that authors/groups body of work
- While I think the paper justified it's focus on TotalCom well and it notes it isn't an empirical work, I think the experiments, if included, should also look at wall clock time, since this, energy consumption, memory load and  Communication Complexity together are arguably the main Figures of Merit in practice, and it is easy to simulate a few representative scenarios with different worker behaviour/latency, as was done in e.g. https://arxiv.org/abs/2210.10311
- More of a question than a weakness: if I understand correctly, you assume a fully reliable federated learning setup where a worker is selected fully by the server and does not drop out under any circumstances once selected? How would unreliable or adversarial workers affect your algorithm?

---

> ### Author Response · Authors · 2023-11-02
> **Response 1/2**
>
> Thank you for your thorough and positive evaluation of our work.
>
> > If i understand correctly, the main "trick" is to exploit the separability of expectation and squared euclidean norm and using an unbiased compression algorithm (top-k). Combined this allows to construct a form of random coordinate descent w.r.t the local objective being executed (by only updating the control variates where the algorithm made progress towards the local minimum), while maintaining global convergence.
>
> Do you mean rand-k? Top-k is biased. It is correct that TAMUNA proceeds "by only updating the control variates where the algorithm made progress towards the local minimum". We are not sure about the analogy with random coordinate descent, though, because compression, probabilistic activation of communication, and partial participation by client sampling form multiple intertwined sources of randomness that we handle in our proof technique. This is reflected by the fact that the proof of Theorem 1 in Appendix A is 8 pages long.
>
> > [Jhunjhunwala et al. "FedVARP: Tackling the Variance Due to Partial Client Participation in Federated Learning", UAI Aug. 2022] seems relevant.
>
> Thank you for mentioning this paper, which we did not know. We have added this reference. The authors propose an interesting SAGA-inspired technique to reduce the variance due to partial participation in Fedavg, in the smooth nonconvex setting, whereas we consider the convex setting. We can note that their assumption 3 on bounded global variance is not compatible with the setting of strongly convex functions with arbitrary heterogeneity on which we focus. For instance, quadratic functions satisfying this assumption must have the exact same quadratic part and differ only by a linear term.
>
> > I think the experiments should also look at wall clock time since this, energy consumption, memory load and Communication Complexity together are arguably the main Figures of Merit in practice
>
> We indeed focus solely on the communication complexity, which is the criterion we aim at minimizing. In fact, the communication bottleneck is the main challenge in federated learning and what makes this field distinctive from classical distributed machine learning. The computation complexity of TAMUNA in number of (stochastic) gradient calls performed in parallel coincides with its iteration complexity and is analyzed in Section 3. Thus, the acceleration from $\kappa$ to $\sqrt{\kappa}$ thanks to local training is present in the communication complexity but not in the computation complexity, as expected. The wall-clock time in experiments includes non-accelerated computation (which itself varies heavily depending on whether full gradients or stochastic gradients are used, and the nature thereof) and accelerated communication, so it is not illustrative of our results. Obtaining computation acceleration would require other techniques, such as Nesterov or momentum-type acceleration, which are notoriously hard to combine with stochastic mechanisms. Moreover, as studied in the paper arXiv:2210.10311 you mention, different clients can have different computation speed and characteristics. The wall-clock time also depends on memory transfers and cache properties, which can vary a lot with respect to the architecture. These are challenging matters addressed by the community of federated learning, which go beyond the scope of our paper.
>
> > you assume a fully reliable federated learning setup where a worker is selected fully by the server and does not drop out under any circumstances once selected?
>
> Yes, in the setting considered presently, the server selects uniformly at random the participating clients. This is a simplistic mathematical model which allows us to analyze the complexity with respect to the number c of participating clients, which is fixed and constant. This model does not really account for unreliable or adversarial workers. It could be generalized to other sampling schemes, for instance importance sampling where different clients have different probabilities of participation. We kept the setting, which is already complex, as simple as possible by considering the same parameter values for all clients.

---

> ### Author Response · Authors · 2023-11-02
> **Response 2/2 - Requested changes**
>
> > adding discussion to the reference I found + further discussion of relevant work.
>
> We added the reference you mentioned. We focus on the scientific question of partial participation with variance reduction, so that linear convergence to the exact solution (with exact gradients) is obtained with arbitrarily heterogeneous functions, while keeping the communication acceleration benefit of local training. We are not aware of papers in this perspective, which we do not cite. Please inform us if we missed any relevant paper.
>
> > the question I would simply be curious about but I think adding it to the discussion would be a mini-strengthening of the work
>
> Sorry but what is the question you are curious about?
>
> > "Telescopic" should probably be "Telescoping"?
>
> Yes, thank you for pointing out this typo.

---

> > ### Comment · Reviewer_7BWA · 2023-11-08
> >
> > "the question" was simply the was referring to the paragraph of "more than a question than a weakness"

---

> > > ### Author Response · Authors · 2023-11-09
> > >
> > > We get it. The topic of asynchronous communication is certainly a better framework to model and accommodate idle or unreliable workers in a robust way.

---

### Comment · Editors_In_Chief · 2023-11-24
**Violation of the dual submission policy**

According to the TMLR submission policy (https://jmlr.org/tmlr/editorial-policies.html), "_[t]here should not be any reuse of written text, figures or results between the submitted paper and any paper which has been published, accepted for publication, or submitted in parallel at another archival, peer-reviewed venue._"

We were made aware of another paper, whose authors overlap significantly with the authors of this submission, was submitted to another peer-reviewed archival venue, with a substantial reuse of text. This submission therefore should be desk-rejected.

The submission was however already reviewed and discussed by the reviewers and action editor by the time we were made aware of this violation. We thus approve the rejection recommendation by the AE and leave this separate note on the violation of the policy.

---

> ### Author Response · Authors · 2023-12-08
> **This decision is unjustified**
>
> We did not violate the dual submission policy, for the following reasons.
>
> The paper you are referring to is an unpublished paper put as a preprint on arXiv, which introduces the CompressedScaffnew algorithm. It was written in October 2022 and revised in January 2023. The present paper introduces TAMUNA, a different algorithm. It was written in May 2023 and submitted to TMLR in August 2023. The latter is a follow-up work of the former, as clearly written in the paper (see Section 1.4.2).
>
> TAMUNA has the same local training and compression mechanisms as CompressedScaffnew, but allows for partial participation. This generalization was highly nontrivial to obtain and took several months of work to achieve. Our paper on TAMUNA stands on its own and we insist that it is not an extended version of the paper on CompressedScaffnew.
>
> Our paper was submitted in August 2023, and at this time the older paper on CompressedScaffnew was not submitted anywhere. Hence, we did not violate any policy when submitting our paper to TMLR.
>
> It turns out that the paper on CompressedScaffnew which, again, is an older work, was submitted to ICLR several months later. There is no reason why its authors would not try to get their work published.
>
> It is true that there is reuse of text in our paper from the previous paper on CompressedScaffnew. This identical text is in the introduction, to present the context and problem of federated learning, and in the literature review. There is no overlap in the description of our contributions and results, which form the substance of the paper. This reuse of text is unfortunate, but it is just an oversight that can be easily fixed by a minor revision. You just had to bring this point to our attention.
>
> A desk-rejection without prior notice and this message casting suspicion are not an appropriate way of dealing with papers submitted to this journal, or to any other venue.

---

> > ### Comment · Editors_In_Chief · 2023-12-11
> >
> > As the authors agree, there is a significant reuse of text from the other paper (on CompressedScaffnew), which was submitted in parallel (i.e., the review processes were overlapping) to ICLR, another archival, peer-reviewed venue. These are the facts of the matter, and as per the text of TMLR's policy quoted above, this violates the dual submission policy. We do not cast any further suspicion beyond pointing this out.

---

### Decision · Action_Editor_ZLFg · 2023-11-21

**Recommendation:** Reject

**Comment:**

**After-rebuttal and author/reviewer discussion**

The reviewers believe that, while the authors addressed part of their concerns, the limited experiments conducted are still a limiting factor towards publication. The reviewers agree that the theoretical results can be provided for a simple convex optimization setting, but deep learning experiments need to be added to have a broader impact on the machine learning community.

This comment should be taken as a universal consensus among the reviewers: they all appreciated the theoretical results and believe the paper is close to publication with some refinements. Yet, reviewers also believe this work reads a bit ``thin'' even after responses compared to CompressedScaffnew (this work only adds partial participation).

As mentioned above, while TMLR's criterion for acceptance is for authors to support their claims, the fact that the authors decide not to test their method on settings that make FL interesting to most of the community leads to the proposed rejection at this point.

**Audience:**

While it is the key difference between TMLR and other venues (that TMLR requires authors to support their claims and just that), the lack of experiments (that are traditionally used in federated learning, such as neural networks) makes this work less interesting in the findings of this paper. This is one of the reasons why this paper is rejected.

**Claims And Evidence:**

The claims and evidence in the submission are supported by accurate and clear evidence.

**Resubmission Of Major Revision:**

The authors may consider submitting a major revision at a later time.